# NeuroCLUS: A Foundation Model with Functional Clustering for Intracranial Neural Decoding

**Hui Zheng** [1]    **Hai-Teng Wang** [1]

## Abstract

Foundation models for intracranial neural recordings aim to learn generalizable representations from large-scale unlabeled data. However, existing approaches rely on suboptimal tokenization schemes – treating individual electrode channels as independent tokens or aggregating them into a single brain-wide representation – which fail to capture the brain's inherent functional modularity. We introduce NeuroCLUS, a foundation model that learns to represent neural activity through data-driven functional clusters. NeuroCLUS is built on a novel two-stage pre-training framework. First, a spatial-temporal model learns a functional context graph between channels via a functional context prediction task. Second, this graph guides a soft clustering of channels into a set of learnable prototype tokens, enabling the transformer backbone to process coherent functional units rather than raw channels. Evaluated across a diverse range of decoding paradigms – including speech perception, speech production, and seizure detection – NeuroCLUS consistently achieves state-of-the-art performance. The discovered functional clusters align with established neurophysiology and offer enhanced interpretability. Our work demonstrates that explicitly modeling functional neural groupings significantly improves the efficiency, generalization, and interpretability of foundation models for intracranial decoding.

## 1. Introduction

Intracranial neural signals, captured via invasive techniques such as intracranial electroencephalography (iEEG), provide a high-fidelity window into brain activity. These signals are pivotal for advancing both fundamental neuroscience and clinical brain-computer interfaces (BCIs), enabling applications ranging from speech decoding (Angrick et al., 2021; Wang et al., 2023; Zheng et al., 2025; Chau et al., 2024) to neurological disorder detection (Yuan et al., 2023; Zhang et al., 2023; Li et al., 2025). Recently, the advent of foundation models for intracranial neural recordings (Wang et al., 2023; Zhang et al., 2023; Yuan et al., 2024; Chau et al., 2024; Oganesian et al., 2025; Carzaniga et al., 2025) has promised a paradigm shift, aiming to learn generalizable neural representations through self-supervised pre-training on large-scale unlabeled data. However, a critical question remains: what constitutes the optimal unit of representation for modeling the brain's functional architecture in these models (Chau et al., 2024; Oganesian et al., 2025)?

Prior foundational approaches have primarily relied on channel-level tokenization (Wang et al., 2023; Zhang et al., 2023; Yuan et al., 2024; Carzaniga et al., 2025), treating each electrode as an independent token. While scalable, this scheme fails to encourage the formation of region-level representations, which have been shown to be crucial for decoding complex cognitive tasks like speech production (Zheng et al., 2025). Alternatively, methods like the Population Transformer (PopT) (Chau et al., 2024) aggregate information from all channels into a single brain-level [CLS] token, while ignoring explicit functional clustering. The BaRISTA model (Oganesian et al., 2025) incorporates anatomical prior by grouping channels based on predefined brain atlas regions. Yet, neuroanatomical boundaries are often imperfect proxies for functional modules, which are defined by coordinated neural activity rather than spatial proximity alone. Consequently, a representation strategy that can autonomously discover and leverage functionally coherent neural clusters from data is still lacking, limiting the efficiency and interpretability of current foundation models.

To bridge this gap, we introduce NeuroCLUS, a foundation model that learns functional-cluster-informed representations for intracranial neural decoding. Our method is driven by a key neuroscientific principle: functionally related neural populations, which may be spatially distributed, exhibit correlated activity patterns. NeuroCLUS realizes this through a novel two-stage pre-training framework. In

[1]Independent Researcher. Correspondence to: Hui Zheng <icml2026.neuroclus@gmail.com>.

the first stage, we pre-train a spatial-temporal model with a spatial context prediction task, learning a functional dependency graph between channels without labeled data. This graph captures how channels influence each other's activity beyond mere anatomical adjacency. In the second stage, we leverage this graph to guide a functional clustering process. Channels are softly assigned to a set of K learnable prototype tokens (e.g., K=8), effectively compressing information into functional region-level tokens. This cluster-aware tokenization allows the transformer backbone to model interactions between coherent functional units rather than raw, potentially noisy channels.

We rigorously evaluate NeuroCLUS across a diverse suite of intracranial decoding tasks, demonstrating its broad utility and superiority. NeuroCLUS achieves state-of-the-art performance on: (1) speech perception tasks in the Brain Treebank (Wang et al., 2024a), (2) the speech production task in the Du-IN dataset (Zheng et al., 2025), and (3) seizure detection tasks in SWEC & MAYO & FNUSA datasets (Carzaniga et al., 2025; Nejedly et al., 2020). Its consistent gains highlight the benefits of functional-cluster-informed representations.

In summary, our contributions are threefold:

- **A Functional-Cluster-Informed Foundation Model:** We propose NeuroCLUS, the first foundation model that explicitly learns to tokenize intracranial neural signals into data-driven functional clusters via a two-stage functional-context-guided pre-training strategy.

- **Function-Guided Representation Learning:** Moving beyond anatomical heuristics, NeuroCLUS discovers functional modules based on estimated inter-channel dependency, yielding representations that better reflect the brain's functional organization.

- **State-of-the-Art & Generalizable Performance:** NeuroCLUS establishes new SOTA results across multiple decoding paradigms (perception, production, detection) and datasets, demonstrating strong generalization across decoding tasks based on iEEG recordings.

## 2. Related Work

### 2.1. Spatiotemporal Modeling for iEEG

Effective modeling of intracranial data requires jointly capturing its rich spatial and temporal structure. Early foundation models primarily focused on temporal dynamics, treating each channel independently (Wang et al., 2023). Subsequent approaches incorporated spatial context, ranging from encoding single-channel coordinates (Chau et al., 2024; Mentzelopoulos et al., 2024) to aggregating signals within pre-defined channel ensembles (Zheng et al., 2025).

Recent models have introduced more sophisticated inductive biases: BaRISTA (Oganesian et al., 2025) integrates a fixed multi-scale neuroanatomical prior, while MVPFormer (Carzaniga et al., 2025) employs a multi-variate parallel attention mechanism to capture interactions across multiple signal sources simultaneously.

Despite their diversity, a common limitation persists: these methods rely on static, pre-defined priors to structure spatial interactions—whether an anatomical atlas or a fixed attention pattern. This assumes the optimal neural groupings for decoding align with known anatomy or uniform mixing weights. However, functionally coherent neural populations are often dynamic and task-specific, not always conforming to rigid anatomical boundaries. Consequently, existing models lack a mechanism to autonomously discover data-driven functional assemblies that reflect the latent organization of neural computation. NeuroCLUS addresses this gap by learning an adaptive functional clustering directly from neural activity, moving beyond static spatial heuristics (Oganesian et al., 2025; Zheng et al., 2025) to uncover the data's inherent functional architecture.

### 2.2. Self-Supervised Modeling for Neural Signals

Self-supervised learning (SSL) has emerged as the dominant paradigm for building foundation models across neural modalities, including iEEG (Wang et al., 2023; Zhang et al., 2023; Chau et al., 2024; Oganesian et al., 2025; Carzaniga et al., 2025), neural spike (Ye et al., 2023; 2025), EEG (Jiang et al., 2024; Wang et al., 2024c;b), etc. For intracranial recordings, a common SSL approach is masked signal reconstruction, where random segments of channel-level time-series are predicted (Wang et al., 2023). While effective for learning local features, this strategy treats channels as independent units, failing to explicitly model the inter-channel dependencies that underlie population-level neural computations. Subsequent spatiotemporal models (Zhang et al., 2023; Carzaniga et al., 2025) attempt to capture these dependencies but often still operate on channel-level tokens, which can limit their ability to form the region-level representations crucial for complex cognitive decoding (Zheng et al., 2025).

To encourage region-level representations, recent work has explored two main pathways. The Population Transformer (PopT) (Chau et al., 2024) aggregates information from all channels into a (brain-level) classification token, improving performance on tasks such as speech perception, while ignoring explicit functional clustering. Alternatively, BaRISTA (Oganesian et al., 2025) incorporates anatomical prior knowledge by grouping channels based on a brain atlas for region-level masking. However, this method relies on static, pre-defined heuristics, which may not align with dynamic functional modules. Our work bridges this gap.

**(a) Functional Context Prediction**

**(b) Pre-training NeuroCLUS**

*Figure 1.* **Overview of NeuroCLUS framework. (a).** Functional Context Pre-training. We train a channel-level iEEG foundation model across multiple subjects to support general context extraction. **(b).** Pre-training NeuroCLUS. We train a region-level iEEG foundation model across multiple subjects to support general functional clustering and downstream neural decoding.

Instead of using fixed anatomical rules or a single aggregated token, NeuroCLUS learns to discover functionally coherent clusters directly from data, enabling the model to dynamically form the optimal units for representation and decoding.

## 3. Method

In this section, we detail the whole framework of Neuro-CLUS, which comprises two pre-training stages (Figure 1). We first formulate the multi-channel iEEG signals as $\mathcal{X} \in \mathbb{R}^{C \times T}$, where $C$ is the number of iEEG channels and $T$ is the total timestamps.

### 3.1. Functional Context Prediction

Prior to pre-training NeuroCLUS (functional clustering modeling), we need to extract the functional context from raw iEEG signals. We introduce a self-supervised objective called functional context prediction, which is trained by discriminating replaced functional contexts in the latent token space (Assran et al., 2023). The architecture consists of several key components: (1) Patch Tokenizer, which segments raw iEEG signals into local patches and projects them into token space; (2) Temporal & Spatial Transformer, which jointly models dependencies across

timestamps and channels to produce contextualized representations. Subsequently, Token Predictor transforms these representations into token space, where a contrastive loss is computed against the patch tokens generated by Target Patch Tokenizer – a momentum-updated version of Patch Tokenizer via Exponential Moving Average (EMA). This design encourages the model to learn meaningful functional contexts by predicting which tokens have been replaced, thereby capturing the underlying structure of multi-channel iEEG activity without requiring supervision labels.

**Model Architecture.** Our model architecture and tokenization scheme are shown in Figure 1 (a). Given the multivariate iEEG signals $\mathcal{X} \in \mathbb{R}^{C \times T}$, we first tokenize channels as univariate signals (i.e., agnostic to space), following common practice (Zhang et al., 2023; Chau et al., 2024; Jiang et al., 2024). We create temporal patches of each channel that are of length $L$ (e.g., 200 milliseconds), yielding $\mathcal{X}_p = \{\boldsymbol{x}_{i,j}^p \in \mathbb{R}^L | i = 1, ..., N; j = 1, ..., C\}$, where $N = \lfloor \frac{T}{L} \rfloor$, the number of patches is $|\mathcal{X}_p| = N \times C$, and $\boldsymbol{x}_{i,j}^p \in \mathbb{R}^L$ indicates the $i$-th patch of length $L$ for the $j$-th channel.

Patch Tokenizer consists of a stack of convolution blocks. In the first step of tokenization, each patch $\mathcal{X}_{i,j}^p$ is passed through Patch Tokenizer (shared across patches). In practice

this tokenizer can take any form; here we choose a temporal convolution neural network (CNN) both to account for the input signal's continuous nature and because of prior domain knowledge about the importance of oscillatory features in neural activity (Jacobs & Kahana, 2010; Buzsaki & Draguhn, 2004). In each convolution block, the temporal convolution layer is stacked with group normalization (Wu & He, 2018), and Gaussian Error Linear Unit (GELU) activation (Hendrycks & Gimpel, 2016). We denote the patch embeddings from Patch Tokenizer as

$$\mathcal{E}_p = \{\boldsymbol{e}_{i,j}^p \in \mathbb{R}^d | i = 1, ..., N; j = 1, ..., C\}, \quad (1)$$

where $d$ is the dimension of embeddings.

To equip the model with awareness of temporal order among patch embeddings, we adopt the parameter-free sinusoidal positional encoding (Vaswani et al., 2017), denoted as $\mathcal{E}_t = \{\boldsymbol{e}_1^t, ..., \boldsymbol{e}_{t_{max}}^t\}$. Here, $t_{max}$ is a hyperparameter that defines the maximum number of time patches and satisfies $t_{max} \geq N$. We choose sinusoidal encodings over learnable alternatives for three primary reasons. First, extrapolation capability: sinusoidal encodings can handle input sequences of arbitrary length, which is essential for accommodating the variable segment durations encountered across different pre-training datasets. Second, parameter efficiency: they introduce no additional trainable parameters, thereby reducing the risk of overfitting. Third, inductive bias: their smooth positional prior aligns naturally with the temporal continuity inherent in neural signals. Given one arbitrary patch embedding $\boldsymbol{e}_{i,j}^p$ in Equation 1, we add the corresponding temporal embedding to it:

$$\mathcal{E} = \{\boldsymbol{e}_{i,j}^p + \boldsymbol{e}_j^t | i = 1, ..., N; j = 1, ..., C\}, \quad (2)$$

which forms the input embeddings $\mathcal{E}$ for the Temporal Transformer. Then, the embeddings will be directly fed into the Transformer encoder (Vaswani et al., 2017) to get the temporal-transformed embeddings $\mathcal{E} = \{\boldsymbol{e}_{i,j} | i = 1, ..., N; j = 1, ..., C\}$.

The functions supported by the same anatomical brain regions in different subjects are roughly similar (e.g., the superior temporal gyrus (STG) consistently participates in speech perception), although fine-grained functional sub-organization may vary between subjects (Buzsáki, 2006). As previous studies demonstrate that injecting anatomical priors leads to no significant performance gap (Mentzelopoulos et al., 2024) and correlations among channels provide inherent relative locations (Zhang et al., 2023), the temporal-transformed embeddings are directly processed by the Spatial Transformer to produce the spatial-transformed embeddings $\mathcal{E}$.

**Functional Context Prediction.** To encourage the model to effectively capture general functional context, we train

the model using a functional context prediction task (Figure 1 (a)). During pre-training, for each iEEG sample $\mathcal{X} \in \mathbb{R}^{C \times T}$, 10% of channels are randomly selected to have their activity replaced by activity from unrelated time points. To ensure balanced label distribution, we designate only 10% of unreplaced channels as positive samples during pre-training. The modified sample is directly fed into the Patch Tokenizer to get the patch embeddings $\mathcal{E}_p$. Then, the patch embedding $\mathcal{E}_p$ is directly fed into the spatial-temporal Transformer to get the transformed embeddings $\mathcal{E}$. While the patch embeddings $\mathcal{E}_p$ are obtained using our original Patch Tokenizer (left part in Figure 1 (a)), we use a separate Target Patch Tokenizer (right part in Figure 1 (a)) for the target embeddings $\hat{\mathcal{E}}_p$ to provide self-supervision signals. The Target Patch Tokenizer is updated with an exponential moving average (EMA) of the original Patch Tokenizer weights. To encourage the model to learn functional context, the model is trained to discriminate replaced channels based on the spatial-temporal transformed embeddings $\mathcal{E} \in \{\boldsymbol{e}_{i,j} | i = 1, ..., N; j = 1, ..., C\}$, the target patch embeddings $\hat{\mathcal{E}}_p \in \{\hat{\boldsymbol{e}}_{i,j}^p | i = 1, ..., N; j = 1, ..., C\}$ and the replaced label $y \in \{-1, 1\}$, where $-1$ indicates unreplaced, $1$ indicates replaced. The functional context loss is defined as follows:

$$\mathcal{L}_{ctx} = \sum_{i,j} \left[1 - y \cdot \langle \ell_2(\text{Linear}(\boldsymbol{e}_{i,j})), \ell_2(\hat{\boldsymbol{e}}_{i,j}^p) \rangle \right], \quad (3)$$

where $\ell_2$ represents $\ell_2$ normalization and $\langle \cdot, \cdot \rangle$ the inner product. Combined with $\ell_2$, $\langle \cdot, \cdot \rangle$ calculates the cosine similarity between $\boldsymbol{e}_{i,j}$ and $\hat{\boldsymbol{e}}_{i,j}^p$, which takes value within [-1,1]-range. $y$ is used to adjust whether to minimize or maximize such similarity, and the shift item $1$ is added to ensure $\mathcal{L}_{ctx} \geq 0$. Although our functional context prediction objective resembles PopT's ensemble discriminative objective (Chau et al., 2024), their purposes differ. PopT targets general neural representation learning, whereas ours focuses on capturing spatial dependencies among channels to provide reliable connectivity priors for clustering, not directly used for downstream decoding or other tasks.

### 3.2. Pre-training NeuroCLUS

Although the above model learns to extract general functional context from raw iEEG signals, it does not explicitly encourage region-level representations (Chau et al., 2024; Zheng et al., 2025), which may limit its effectiveness for cognitive decoding tasks (e.g., speech perception and production). A promising way to capture such region-level structure is to introduce learnable cluster prototypes. However, the traiditional reconstruction loss (Zhang et al., 2023; Jiang et al., 2024) alone cannot meaningfully update or constrain these prototypes – without an additional inductive bias, the gradient from reconstruction provides no incentive for the model to form coherent clusters. To address this, we

propose a functional clustering modeling framework (Figure 1 (b)) that couples reconstruction with a dedicated clustering objective. The key components are (1) Patch Tokenizer, (2) Spatial & Temporal Transformer, (3) Cluster Prototypes, which collectively encode iEEG samples into both channel-level and region-level representations. The channel-level representations are trained to reconstruct the original signal via a reconstruction loss, preserving fine-grained neural information. The region-level representations, in contrast, are optimized by a clustering loss that leverages functional connectivity priors; this loss compresses functionally similar channels into compact prototypes, thereby enabling the model to learn meaningful region-level abstractions that would otherwise be inaccessible from reconstruction alone.

**Model Architecture.** Our model architecture and tokenization scheme are shown in Figure 1 (b). Given iEEG signals $\mathcal{X} \in \mathbb{R}^{C \times T}$, we create temporal patches of each channel, yielding $\mathcal{X}_p = \{\boldsymbol{x}_{i,j}^p \in \mathbb{R}^L | i = 1, ..., N; j = 1, ..., C\}$.

Patch Tokenizer transforms patches into patch tokens $\mathcal{E}_p$. To capture region-level representations, we initialize a set of $K$ cluster tokens $\mathcal{C} = \{\boldsymbol{c}_k \in \mathbb{R}^d | k = 1, ..., K\}$. Given the patch tokens $\mathcal{E}_p$, we concatenate cluster tokens position-wise to formulate the initial embeddings $\mathcal{E}_{\text{init}} = \{\boldsymbol{e}_{i,j} \in \mathbb{R}^d | i = 1, ..., N; j = 1, ..., C + K\}$. Then, the embeddings are fed into Spatial & Temporal Transformer to get the spatiotemporal-transformed embeddings $\mathcal{E}$. Finally, the embeddings are fed into the decoder to reconstruct the neural signals $\hat{\mathcal{X}}_p = \{\hat{\boldsymbol{x}}_{i,j}^p \in \mathbb{R}^L | i = 1, ..., N; j = 1, ..., C\}$, thus preserving the information. The mean square error (MSE) loss is:

$$\mathcal{L}_{mse} = \frac{1}{N \cdot C} \sum_{i,j} ||\boldsymbol{x}_{i,j}^p - \hat{\boldsymbol{x}}_{i,j}^p||_2^2. \qquad (4)$$

**Functional Clustering Modeling.** To encourage Neuro-CLUS to capture region-level representations, we train NeuroCLUS using a functional clustering modeling task. Specifically, we pass all iEEG recordings of each subject through the functional context extractor and extract the attention matrices from the spatial Transformer. These matrices are then averaged across layers, temporal patches, and samples to obtain a stable sparse connectivity graph $\mathcal{P} \in \mathbb{R}^{C \times C}$ per subject, which serves as a prior for channel clustering. This graph is subsequently broadcast across all $N$ time steps, providing a self-supervision signal during pre-training NeuroCLUS. Accordingly, the probability that a given $j$-th channel is associated with the $k$-th cluster is the normalized inner-product of the cluster-level embeddings and the channel-level embeddings, which is computed as

$$p_{i,j,k} = \text{Normalize}\left(\frac{\boldsymbol{e}_{i,k}^{r\top} \boldsymbol{e}_{i,j}^c}{||\boldsymbol{e}_{i,k}^r|| \cdot ||\boldsymbol{e}_{i,j}^c||}\right) \in [0, 1], \qquad (5)$$

where $\boldsymbol{e}_{i,k}^r = \boldsymbol{e}_{i,C+k}$ and $\boldsymbol{e}_{i,j}^c = \boldsymbol{e}_{i,j}$. The normalization operator ensures that $\sum_k p_{i,j,k} = 1$ and validates the clustering probability distribution across $k$ clusters. We utilize the reparameterization trick (Jang et al., 2016) to obtain the clustering membership matrix $\mathcal{M} \in \mathbb{R}^{N \times C \times K}$, where $\mathcal{M} = \text{Bernoulli}(p_{i,j,k})$. Higher probability $p_{i,j,k}$ results in $\mathcal{M}_{i,j,k}$ close to 1, leading to the deterministic existence of certain channels in the corresponding cluster.

We further introduce a specifically designed loss function $\mathcal{L}_{clus}$ for the clustering quality, which incorporates both the alignment of channels with respective clusters and the distinctness between different clusters in a self-supervised context. The cluster loss $\mathcal{L}_{clus}$ is formulated as:

$$\mathcal{L}_{clus} = \frac{1}{N} \sum_{i=1}^N \Big[ -\text{Tr}\left(\mathcal{M}_i^\top \mathcal{P}_i \mathcal{M}_i\right) \\ + \text{Tr}\left(\left(1 - \mathcal{M}_i \mathcal{M}_i^\top\right) \mathcal{P}_i\right) \Big], \qquad (6)$$

where Tr indicates a trace operator. $\text{Tr}(\mathcal{M}_i^\top \mathcal{P}_i \mathcal{M}_i)$ maximizes the channel similarities within clusters, which is a fundamental requirement for effective clustering. $\text{Tr}((1 - \mathcal{M}_i \mathcal{M}_i^\top)\mathcal{P})$ instead encourages separation between clusters, which further prevents overlap and ambiguity in clustering assignments. $\mathcal{L}_{clus}$ captures meaningful region-level prototypes without relying on external labels or annotations. The overall loss function thereby becomes $\mathcal{L} = \mathcal{L}_{mse} + \mathcal{L}_{clus}$. Notably, NeuroCLUS employs learned prototypes at every time step, and only the updated prototypes (i.e., region-level representations) are used for downstream classification. When per-timestamp embeddings are required, we concatenate prototype embeddings across prototypes. When per-sequence embeddings are needed, we concatenate per-timestamp embeddings across time steps.

## 4. Experiments

### 4.1. Evaluation Datasets

We systematically evaluate our NeuroCLUS on the following downstream datasets:

- **SWEC dataset** (seizure detection) (Carzaniga et al., 2025): A corpus of iEEG recordings with a sampling rate of either 512Hz or 1024 Hz, containing 68 subjects for a total of 9,328 hours. All the recordings were inspected by an expert for identification of seizure onsets and offsets. We strictly adhere to the evaluation protocol established in MVPFormer (Carzaniga et al., 2025): training on the same 18 subjects and testing in a zero-shot manner on unseen 50 subjects.

- **MAYO Dataset** (seizure detection) (Nejedly et al., 2020): A corpus of iEEG recordings with a sampling

rate of 32kHz, containing 24 subjects for a total of 130 hours. All the recordings were inspected by an expert for the identification of ictal events. We strictly adhere to the evaluation protocol established in MVPFormer (Carzaniga et al., 2025): training on the first 4 subjects and testing in a zero-shot manner on the remaining unseen subjects.

- **FNUSA Dataset** (seizure detection) (Nejedly et al., 2020): A corpus of iEEG recordings with a sampling rate of 25kHz, containing 14 subjects for a total of 160 hours. All the recordings were inspected by an expert for the identification of ictal events. We strictly adhere to the evaluation protocol established in MVPFormer (Carzaniga et al., 2025): training on the first 4 subjects and testing in a zero-shot manner on the remaining unseen subjects.

- **Brain Treebank** (speech perception) (Wang et al., 2024a): A corpus of iEEG recordings with a sampling rate of 2048Hz, containing 10 subjects for a total of 55 hours. All the recordings were collected during the movie-watching task, including 4 classification tasks (i.e., pitch, volume, sentence onset, and speech/non-speech). We strictly adhere to the evaluation protocol established in PopT (Chau et al., 2024): (1) selecting the Top-90 most informative channels, and (2) performing hold-out cross-validation within each subject.

- **Du-IN dataset** (speech production) (Zheng et al., 2025): A corpus of iEEG recordings with a sampling rate of 1000Hz, containing 12 subjects for a total of 36 hours. All recordings were collected during the word-production task, and the word set included 61 Chinese words. We strictly adhere to the evaluation protocol established in Du-IN (Zheng et al., 2025): (1) selecting the Top-10 most informative channels to exclude channels from irrelevant brain regions, and (2) performing hold-out cross-validation within each subject.

### 4.2. Experiment Setup

**Pre-training & Fine-tuning.** For pre-training Neuro-CLUS and the functional context prediction, we collect a total time of ∼10k hours from public datasets; see Appendix A for more details. Following common practice in foundation model evaluation, the data for the five downstream tasks were included in the pre-training corpus; we performed dedicated ablation studies (Figure 2) to analyze the model's generalization. For each downstream benchmark, we adhered strictly to the official data splits and evaluation protocols established in their respective original works (Carzaniga et al., 2025; Chau et al., 2024; Zheng et al., 2025) to ensure a fair comparison. We employ binary cross-entropy (BCE) loss for the SWEC & MAYO & FNUSA datasets and the Brain Treebank (binary classification) and cross-entropy

loss for the Du-IN dataset (multi-class classification), respectively. Our Experiments are conducted on eight A100 GPUs by Python 3.11.7 and PyTorch 2.0.2 + CUDA 12.3. The best models are trained based on the training set, selected from the validation set, and finally evaluated on the test set. We report the average and standard deviation values on six different random seeds to obtain comparable results.

**Preprocessing.** We first filter the iEEG signals between 0.5Hz and 200Hz to remove low-frequency noise. Then, a notch filter of 50Hz (or 60Hz) is applied to avoid power-line interference. Next, all iEEG signals are resampled to 400Hz and median-referenced. Finally, z-score normalization is performed on each channel to standardize data scales.

**Baselines & Metrics.** The baselines mainly focus on advanced iEEG foundation models (Zhang et al., 2023; Chau et al., 2024; Oganesian et al., 2025), dataset-paired baselines (Carzaniga et al., 2025; Zheng et al., 2025) and other advanced neural decoding baselines (Jiang et al., 2024; Wu et al., 2024). All baselines were pre-trained from scratch on collected datasets before downstream fine-tuning, rather than initialized from provided checkpoints; see Appendix C for more details. We use the following metrics for comparison: (1) **Balanced Accuracy**: the average of recall on each class, which is utilized for multi-class classification; (2) **ROC-AUC**: area under the receiver operating characteristic curve, which is used for binary classification; (3) **Cohen's Kappa**: a measure of agreement between categorical variables $\mathcal{X}$ and $y$, which is calculated from the observed and expected frequencies on the diagonal of a square contingency table. It is used for binary classification. (4) **F1-Score**: A harmonic mean of the precision and recall, where the relative contribution of precision and recall to the F1 score is equal, which is used for binary classification.

### 4.3. Comparison with Baselines

NeuroCLUS establishes new state-of-the-art performance across multiple iEEG decoding paradigms, as summarized in Table 1. In the context of seizure detection, evaluated on three clinical datasets (SWEC (Carzaniga et al., 2025), MAYO, and FNUSA (Nejedly et al., 2020)), our method attains the highest Cohen's Kappa and F1 scores among all competing approaches, demonstrating that its learned functional clusters are highly discriminative for pathological states. Turning to speech perception on the Brain Treebank benchmark (Wang et al., 2024a), NeuroCLUS again delivers substantial overall improvements, with particularly notable gains on the more challenging higher-level tasks of sentence onset detection and speech/non-speech discrimination. Taken together, these findings suggest that modeling interactions among coherent, data-driven functional units – rather than treating isolated channels or relying on fixed

*Table 1.* Results on iEEG datasets. We compare NeuroCLUS with multiple baselines across 5 iEEG datasets and 6 tasks. The best results are **bolded**, while the second results are underlined.

| Methods | SWEC | | MAYO | FNUSA | Brain Treebank | | | | Du-IN |
|---|---|---|---|---|---|---|---|---|---|
| | Seizure | | Seizure | Seizure | Pitch | Volumn | Onset | Speech | Word |
| | Kappa | F1 | F1 | F1 | ROC-AUC | ROC-AUC | ROC-AUC | ROC-AUC | BAC (%) |
| Brant | 0.41±0.03 | 0.39±0.03 | 0.27±0.02 | 0.42±0.03 | 0.61±0.04 | 0.74±0.03 | 0.80±0.03 | 0.80±0.03 | 12.42±4.10 |
| LaBraM | 0.51±0.02 | 0.47±0.02 | 0.33±0.02 | 0.44±0.03 | 0.72±0.02 | 0.86±0.02 | 0.89±0.02 | 0.94±0.01 | 12.82±2.51 |
| Du-IN | 0.53±0.02 | 0.49±0.02 | 0.32±0.02 | 0.45±0.02 | 0.77±0.02 | 0.87±0.02 | 0.90±0.02 | 0.91±0.02 | 62.70±4.69 |
| H2DiLR | 0.52±0.02 | 0.47±0.02 | 0.30±0.02 | 0.43±0.03 | 0.74±0.02 | 0.87±0.02 | 0.88±0.02 | 0.90±0.02 | 23.92±3.79 |
| PopT | 0.54±0.02 | 0.51±0.02 | 0.34±0.01 | 0.43±0.02 | 0.74±0.03 | 0.87±0.03 | 0.90±0.01 | 0.93±0.02 | 42.57±3.80 |
| BaRISTA | 0.54±0.02 | 0.52±0.03 | 0.30±0.02 | 0.45±0.02 | 0.73±0.03 | 0.85±0.03 | 0.91±0.02 | 0.92±0.02 | 18.94±3.98 |
| MVPFormer | 0.57±0.02 | 0.56±0.03 | 0.36±0.02 | 0.46±0.02 | 0.83±0.02 | 0.88±0.02 | 0.87±0.03 | 0.90±0.02 | 15.74±3.66 |
| NeuroCLUS | **0.61±0.01** | **0.59±0.02** | **0.40±0.02** | **0.51±0.02** | **0.83±0.02** | **0.92±0.02** | **0.96±0.01** | **0.99±0.01** | **65.92±4.17** |

anatomical groupings – produces neural representations that more faithfully capture the integrated activity underlying complex perceptual processes.

The key advantage of our functional clustering approach becomes most apparent on the challenging Du-IN speech production task. Prior foundation models that rely on channel-level tokenization (Brant, 12.42%; LaBraM, 12.82%) or brain-level tokenization (PopT, 42.57%) fail to learn meaningful region-level representations, leading to a substantial performance gap relative to the strong task-specific Du-IN baseline (62.70%). Notably, while both H2DiLR and Du-IN aggregate channels within pre-defined brain regions to produce region-level tokens, H2DiLR replaces continuous embedding sequences with discrete code sequences, which causes a dramatic drop in performance (62.70% vs. 23.92%). NeuroCLUS directly addresses this limitation by explicitly grouping channels into data-driven functional clusters, rather than relying on fixed anatomical boundaries. This design enables the model to learn coherent, interpretable region-level representations that better reflect the underlying neural organization. As a result, NeuroCLUS outperforms all baseline methods – including the specialized Du-IN model – with a balanced accuracy of 65.92%. Together, these results validate our core hypothesis: learning functional modularity directly from neural data is not merely beneficial but essential for building neural decoding models that are both generalizable and efficient.

### 4.4. Pre-training with/without Downstream Datasets

A core objective of our pre-training framework is to learn generalizable iEEG representations that transfer effectively to unseen subjects and tasks, rather than memorizing dataset-specific features. To rigorously evaluate this property, we conducted an ablation study examining how the inclusion of downstream-task data in the pre-training corpus affects model performance. Given that the SWEC dataset makes up the vast majority (approximately 95%) of our pre-training data, we focused on ablating the presence of the other downstream datasets – MAYO & FNUSA datasets,

Brain Treebank, and Du-IN dataset. As shown in Figure 2, removing any of these datasets from pre-training does not lead to a significant change in performance on the corresponding downstream tasks. This result provides strong evidence that NeuroCLUS successfully learns universal neural representations that do not rely on prior exposure to the specific task data used during evaluation.

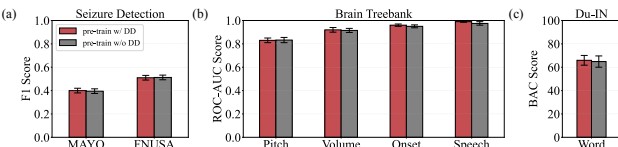

*Figure 2.* **Ablations on whether incorporating downstream datasets into pre-training process. (a).** Model Performance on seizure detection, whether pre-training NeuroCLUS with/without MAYO & FNUSA datasets (290 hours). **(b).** Model Performance on Brain Treebank, whether pre-training NeuroCLUS with/without Brain Treebank (55 hours). **(c).** Model Performance on Du-IN dataset, whether pre-training NeuroCLUS with/without Du-IN dataset (36 hours). Across all comparisons, with the exception of Speech task ($p < 0.05$; paired T-test w/ 6 random seeds) on Brain Treebank, none of differences reached statistical significance.

This finding carries important practical implications. It demonstrates that large-scale, unlabeled iEEG recordings alone are sufficient to pre-train a highly generalizable foundation model, thereby substantially reducing the labeling burden that often constrains supervised learning approaches. It also offers a constructive guideline for future data collection efforts: expanding the volume and subject diversity of unlabeled iEEG corpora is a viable and powerful strategy for building more robust and universally applicable neural decoding models. More broadly, these results reassure practitioners that even when the target task data cannot be included during pre-training – a common scenario in real-world applications – NeuroCLUS maintains strong generalization capabilities, making it a practical and scalable solution for a wide range of neural decoding problems.

## 4.5. Scaling Samples during Pre-training

Despite leveraging a substantial corpus of ∼10k hours of iEEG data from 128 subjects for pre-training, the scale of our dataset remains a primary limitation. The intrinsic sparsity and high inter-subject variability of intracranial electrode placements suggest that the current data volume may not suffice to fully saturate the representational capacity of a foundation model enhanced with functional clustering.

To probe this scaling relationship, we conducted a controlled experiment by progressively increasing the volume of pre-training samples. As illustrated in Figure 3, model performance on downstream tasks shows consistent improvement with more data, although the marginal gains begin to diminish after ∼2k5 hours. This observation aligns with scaling laws observed in other domains (Kaplan et al., 2020), suggesting that performance would likely continue to improve with orders-of-magnitude more data—potentially on the scale of 100k hours or more. Thus, while our results are promising, they likely represent a lower bound on the model's potential. A key open question for the field is determining the optimal data scale for pre-training iEEG foundation models, which future work with larger, more diverse cohorts will help to answer.

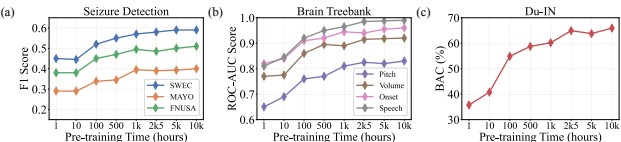

*Figure 3.* **Ablations on total time of iEEG recordings during pre-training process. (a).** Model Performance on seizure detection as the pre-training data increases. **(b).** Model Performance on Brain Treebank as the pre-training data increases. **(c).** Model Performance on Du-IN dataset as the pre-training data increases.

## 4.6. Ablations on Model Architecture

We conduct systematic ablations on model design to verify their effectiveness, as shown in Figure 4.

**Ablations on Functional Context Prediction.** To validate the efficacy of our functional context prediction (FCP) task in estimating the inter-channel connectivity, which provides self-supervision signals for pre-training NeuroCLUS, we conduct an ablation study by replacing it with a conventional Masked Signal Prediction (MSP) task (Zhang et al., 2023). While the MSP task directly reconstructs masked temporal patches and may encourage the model to rely on shortcuts from intra-channel dynamics, our FCP task is explicitly designed to force it to reason about contextual relationships between channels to identify replaced signals.

As shown in Figure 4 (a), the model pre-trained with MSP achieves competitive but consistently and statistically sig-

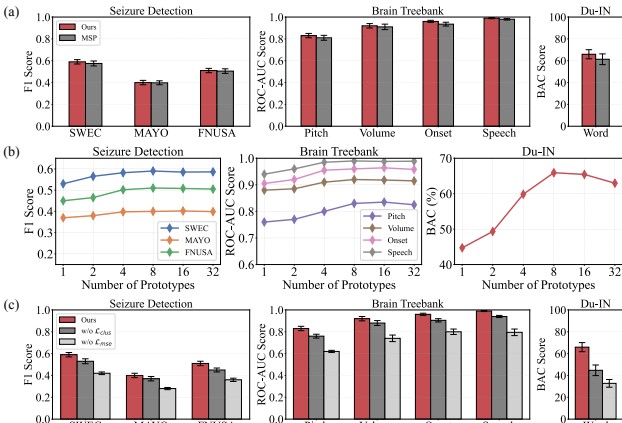

*Figure 4.* **Ablations on model architecture. (a).** Ablations on Functional Context Prediction. MSP means using traditional Masked Signal Prediction for pre-training the functional context extractor. All comparisons show statistically significant differences ($p < 0.01$; paired T-test w/ 6 random seeds). **(b).** Ablations on Prototypes. We vary the number of prototypes to examine its robustness. **(c).** Ablations on Pre-training Loss. We ablate pre-training loss components to verify their effectiveness.

nificantly ($p < 0.01$ with paired T-test) lower performance across all benchmark tasks compared to the FCP variant. This performance gap, though not catastrophic, confirms our hypothesis: while MSP learns useful representations, it is less effective at capturing the functional dependencies that are critical for forming coherent neural clusters and subsequently for superior generalization in downstream decoding. Thus, the FCP task provides a more direct and effective self-supervision signal for understanding the brain's functional connectivity, which is fundamental to our model's success. More ablation results are demonstrated in Appendix I.

**Ablations on Prototypes.** To analyze the impact of functional granularity in NeuroCLUS, we ablate the number of cluster prototypes $K$. As shown in Figure 4 (b), the performance of NeuroCLUS increases substantially from $K = 1$ to $K = 8$, then plateaus as $K$ further increases. This indicates that a minimal set of prototypes is necessary to capture basic functional segregation, but performance saturates once a sufficiently expressive number (around $K = 8$) is reached. This saturation point likely reflects a match between the model's representational capacity and the inherent spatial resolution or functional modularity present in typical iEEG recordings. The stability across $K \in \{8, 16, 32\}$ justifies our choice of $K = 8$ as a computationally efficient and representationally adequate configuration, balancing model complexity with the ability to learn functional clusters.

**Ablations on Pre-training Loss.** We conduct an ablation study on the loss terms used during pre-training Neuro-CLUS to validate their distinct contributions (Figure 4 (c)

& Appendix J). Removing the clustering loss ($\mathcal{L}_{clus}$) prevents the model from forming structured functional modules based on the estimated inter-channel connectivity, leading to a significant performance drop. For instance, on the Du-IN task, BAC falls from 65.92% to 44.72%, confirming that the explicit clustering objective is crucial for learning meaningful region-level representations. Conversely, removing the reconstruction loss ($\mathcal{L}_{mse}$) causes a collapse in the representation space, as the model loses the incentive to preserve fine-grained signal information. This results in uniformly poor performance across all tasks (e.g., Du-IN BAC: 32.70%), demonstrating that $\mathcal{L}_{mse}$ is essential for maintaining informational integrity. The full model, combining both losses, achieves the optimal balance, enabling both effective functional compression and faithful signal representation.

## 5. Limitations

While NeuroCLUS demonstrates strong generalizability, our study has certain limitations. The scale of pre-training recordings, while substantial ($10k$ hours from 128 subjects), may not yet be sufficient to reach the full potential of such a foundation model (enhanced with functional clustering). Given the inherent sparsity and inter-subject variability of iEEG electrode placements, a truly comprehensive model of brain function likely requires orders-of-magnitude more data from a vastly larger and more diverse cohort. The performance gains we observe are promising, but scaling laws in this domain remain unexplored; larger-scale pre-training might yield further improvements in generalization.

Our evaluation, though comprehensive across its scope, is necessarily bounded. We have validated NeuroCLUS on core paradigms within auditory/language processing (speech perception and production) and a major clinical application (seizure detection). However, its performance on other critical neuro-decoding domains – such as visual perception (Liu et al., 2024), motor imagery (Song et al., 2022; Altaheri et al., 2023), or affective computing (Zheng et al., 2017; Hong et al., 2024) – remains unexplored. Assessing the model's generalizability across this wider spectrum of cognitive functions is an essential direction for future work to fully establish its utility as a unified foundation model.

## 6. Discussion

NeuroCLUS demonstrates that learning data-driven functional clusters is a critical advancement for intracranial neural decoding. Its state-of-the-art performance across speech and seizure tasks confirms that cluster-aware tokenization yields more generalizable neural abstractions. The decisive gain on Du-IN speech production specifically validates that forming meaningful region-level representations is essential for decoding complex generative processes. This success

stems from a fundamental shift: rather than relying on independent channels or static anatomical maps, our model directly discovers the brain's latent functional architecture from population activity.

Methodologically, NeuroCLUS reconciles scalability with neuroscientific principle. Prior work presented a trade-off: channel-level models (Zhang et al., 2023; Jiang et al., 2024; Carzaniga et al., 2025) lack regional integration, aggregated models (Chau et al., 2024) lose spefication of different functional regions, and anatomy-guided models (Oganesian et al., 2025) overlook the inconsistency between anatomical and functional regions. Our two-stage framework offers a principled solution. By first learning an inter-channel dependency graph and then using it to guide soft clustering, the model learns to interact with coherent functional units. This yields representations that are computationally efficient and neuroscientifically interpretable.

For neuroscience, NeuroCLUS offers a novel data-driven tool for analyzing large-scale iEEG recordings, enabling the discovery of functional modules that may transcend traditional anatomical boundaries. Further details on the neuroscientific interpretation of our clustering results can be found in Appendix G. For brain-computer interfaces, NeuroCLUS establishes a more robust foundation for critical applications such as speech prostheses and seizure prediction, where reliable decoding of neural activity is paramount. Looking ahead, the functional clustering paradigm introduced here has the potential to extend to other neural recording modalities and to inspire new self-supervised objectives for population-level representation learning. Ultimately, we argue that effective foundation models for neural data must do more than merely process signals; they must actively reason about the brain's functional topology to achieve both high performance and interpretability.

## 7. Conclusion

This paper identifies a fundamental limitation in current foundation models for intracranial neural decoding: their inability to learn functionally coherent, region-level representations from channel-level tokenization. To address this, we introduced NeuroCLUS, a novel foundation model that employs a two-stage pre-training strategy to explicitly discover and leverage data-driven functional clusters. Extensive experiments across speech perception, speech production, and seizure detection tasks demonstrate that NeuroCLUS consistently sets a new state-of-the-art. Crucially, its significant gain on the challenging Du-IN speech production benchmark validates that modeling functional modularity is key to decoding complex cognitive processes. Our work establishes functional clustering as a powerful paradigm for building more effective and brain-inspired foundation models for neural decoding.

## Impact Statement

This research advances the development of foundation models for high-fidelity neural signal decoding. The core methodology and insights from NeuroCLUS can potentially accelerate neuroscience research by providing a data-driven framework for mapping the brain's functional organization. In the clinical domain, improved neural decoding models directly enhance the capabilities of next-generation brain-computer interfaces (BCIs), with significant potential to restore communication for individuals with severe motor impairments and to refine diagnostic tools for neurological disorders like epilepsy.

However, the ability to decode brain activity from invasive recordings raises profound ethical considerations. The privacy and security of highly sensitive intracranial neural data are paramount, and the technology could be misused for purposes beyond therapeutic intent, such as non-consensual inference of cognitive or emotional states. We emphasize that the development and deployment of such models must be guided by strict ethical protocols, including robust data anonymization, informed consent, and transparent communication of the model's limitations. We advocate for ongoing dialogue within the research community to establish governance frameworks that ensure these powerful tools are used responsibly and for broad societal benefit.

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

# A. Dataset Description

We describe the datasets we use for training NeuroCLUS here. The total time across datasets is ∼10k hours from 128 subjects, used for both functional context prediction and pre-training NeuroCLUS. Then, each dataset is used to fine-tune NeuroCLUS for different downstream tasks.

**SWEC Dataset (Carzaniga et al., 2025):** A large-scale iEEG dataset consisting of a total of 68 subjects, 9328 hours of recording, and 704 ictal events. The SWEC iEEG dataset is the largest publicly available iEEG dataset, fully curated and labelled by experienced clinicians. We adopt the same task specification in MVPFormer (Carzaniga et al., 2025), yielding 5-second neural activity per trial. For these classification tasks, we average embeddings and add a linear head after either pre-trained or randomly initialized models. Training employs binary cross-entropy (BCE) loss, with results quantified using Kappa and F1 scores.

**MAYO Dataset (Nejedly et al., 2020):** An iEEG dataset consisting of a total of 24 subjects, 130 hours of recording. The MAYO dataset focuses on seizure detection, which was carefully annotated by experienced clinicians. We adopt the same task specification and training configuration as those in the SWEC dataset.

**FNUSA Dataset (Nejedly et al., 2020):** An iEEG dataset consisting of a total of 14 subjects, 160 hours of recording. The FNUSA dataset focuses on seizure detection, which was carefully annotated by experienced clinicians. We adopt the same task specification and training configuration as those in the SWEC dataset.

**Brain Treebank (Wang et al., 2024a):** An iEEG dataset consisting of a total of 10 subjects, 55 hours of recording. Brain Treebank focuses on speech perception, including four classification tasks (i.e., pitch discrimination, volume discrimination, sentence onset detection, and speech vs. non-speech detection). We adopt the same task specification in PopT (Chau et al., 2024), yielding 5-second neural activity per trial. For these classification tasks, we flatten embeddings and add a linear head after either pre-trained or randomly initialized models. Training employs binary cross-entropy (BCE) loss, with results quantified using ROC-AUC scores.

**Du-IN Dataset (Zheng et al., 2025):** An iEEG dataset consisting of a total of 12 subjects, 36 hours of recording. The Du-IN dataset focuses on speech production, including one challenging classification task (i.e., word classification). We adopt the same task specification in Du-IN (Zheng et al., 2025), yielding 3-second neural activity per trial. For these classification tasks, we flatten embeddings and add a linear head after either pre-trained or randomly initialized models. Training employs binary cross-entropy (BCE) loss, with results quantified using Top-1 accuracy.

# B. Data Augmentation

To enhance the robustness of learned representations, we apply data augmentation in both datasets.

**Pre-training dataset.** In our implementation, we segment neural recordings into 10-second samples with a 5-second overlap. When fetching a sample, we randomly select a starting point between 0 and 5 seconds, then extract a 5-second sample beginning from that point.

**Downstream dataset.** Since trials occur consecutively without gaps, employing the jittering mentioned above leads to the blending of information from other trials. In our implementation, we segment iEEG recordings into samples with the corresponding trial length; see Appendix A for details. When fetching a sample, we randomly choose a shift step between 0 and 0.2 seconds, then shift the sample either to the left or right, padding it with zeros.

## C. Baseline Details

We provide the details of baselines here. The baselines mainly focus on advanced iEEG models (Zhang et al., 2023; Chau et al., 2024; Oganesian et al., 2025), dataset-paired baselines (Zheng et al., 2025; Carzaniga et al., 2025) and other advanced neural decoding baselines (Jiang et al., 2024; Wu et al., 2024).

**Brant (Zhang et al., 2023):** A self-supervised model for iEEG recordings that can capture both long-term temporal dependency and spatial correlation from neural signals. Brant is mainly designed for medicine, serving as an iEEG foundation model. Although Brant mainly evaluates its performance on the low-level modeling tasks (e.g., neural signal forecasting, imputation, etc.), Brant achieves SOTA performance on some high-level modeling tasks (e.g., seizure detection). In practice, we adopt the hyperparameter setting of Brant-Tiny and use all collected iEEG recordings to pre-train it.

**LaBraM (Jiang et al., 2024):** A self-supervised model for EEG recordings that learns generic representations with tremendous EEG data. LaBraM serves as an EEG foundation model, achieving SOTA performance on various downstream EEG tasks. In practice, we adopt the hyperparameter settings of LaBraM-Base and use all collected iEEG recordings to pre-train it. Since the spatial embeddings are pre-defined according to the EEG caps, we replace the learnable spatial embeddings with hard-coded spatial embeddings from PopT (Chau et al., 2024). Furthermore, as most iEEG datasets (Carzaniga et al., 2025; Nejedly et al., 2020; Zheng et al., 2025) do not provide anatomical coordinates for the channels, we skip the spatial encoding step and proceed directly to the subsequent computations.

**Du-IN (Zheng et al., 2025):** A self-supervised model for iEEG-based speech decoding that learns contextual embeddings based on region-level tokens through discrete codex-guided mask modeling. Du-IN achieves SOTA performance on iEEG-based speech decoding using the Du-IN dataset (Zheng et al., 2025). In practice, we adopt the original hyperparameter setting and use iEEG recordings from each subject to pre-train the model.

**H2DiLR (Wu et al., 2024):** A self-supervised model for iEEG-based tone decoding that disentangles and learns both the homogeneity and heterogeneity from intracranial iEEG recordings across multiple subjects. H2DiLR achieves SOTA performance on iEEG-based tone decoding using the iEEG dataset from (Feng et al., 2023). In practice, we adopt the original hyperparameter settings and use all collected iEEG recordings to pre-train it.

**PopT (Chau et al., 2024):** A self-supervised model for iEEG that learns population-level codes for arbitrary ensembles of neural recordings at scale. PopT stacks on top of pre-trained temporal embeddings and enhances downstream decoding by enabling the learned aggregation of multiple spatially sparse channels. PopT serves as an iEEG foundation model, achieving SOTA performance on Brain Treebank (Wang et al., 2024a). In practice, we adopt the original hyperparameter settings and use all collected iEEG recordings to pre-train it. Furthermore, as most iEEG datasets do not provide anatomical coordinates for the channels, we skip the spatial encoding step and proceed directly to the subsequent computations.

**BaRISTA (Oganesian et al., 2025):** A self-supervised model for iEEG that learns flexible representations by independently controlling the spatial scale of channel encoding and masked reconstruction, like GAFormer (Xiao et al., 2024). BaRISTA uses neuroanatomical meta-information (e.g., channel coordinates, atlas parcels, lobes) to guide masking and supports spatial scales beyond individual channels. On Brain Treebank (Wang et al., 2024a), BaRISTA substantially outperforms prior iEEG foundation models, demonstrating that larger-scale spatial encoding critically enhances downstream decoding performance and generalizes to unseen subjects. In practice, we adopt the original hyperparameter settings and use all collected iEEG recordings to pre-train it. Furthermore, as most iEEG datasets do not provide anatomical coordinates for the channels, we skip the spatial encoding step and proceed directly to the subsequent computations.

**MVPFormer (Carzaniga et al., 2025):** A self-supervised model for iEEG that introduces multi-variate parallel attention (MVPA), a self-attention mechanism disentangling content, temporal, and spatial information to handle heterogeneous channel configurations across subjects. Pre-trained using contrastive next-segment prediction with wavelet-based embeddings, MVPFormer achieves expert-level seizure detection, outperforms state-of-the-art baselines on Brain TreeBank decoding tasks, and matches or exceeds leading time-series models on forecasting and classification benchmarks. In practice, we adopt the hyperparameter setting of MVPFormer-M and use all collected iEEG recordings to pre-train it.

## D. Model Details

The functional context pre-training stage (Table 2; 3.39 M) aims to learn a neural context extractor that can extract functional context even from unseen subjects. The functional clustering pre-training stage (Table 3; 5.34 M) aims to learn a neural encoder that aggregates channel-level representations into region-level representations, thereby enabling a wide range of neural decoding tasks (e.g., seizure detection (Carzaniga et al., 2025; Nejedly et al., 2020), speech perception (Wang et al., 2024a), and speech production (Zheng et al., 2025)).

*Table 2.* The hyperparameters for functional context pre-training.

| Module | Sub-Module | Name | Value |
|---|---|---|---|
| Neural Context Extractor | Patch Tokenizer | # of Input Channels | {1,128,128,128} |
| | | # of Output Channels | {128,128,128,128} |
| | | Kernel Size | {9,7,3,3} |
| | | Stride | {5,4,2,2} |
| | | Padding | {4,3,1,1} |
| | | EMA Rate | 0.99 |
| | Temporal Transformer | # of Transformer Layers | 4 |
| | | Hidden Size | 128 |
| | | MLP Size | 512 |
| | | MLP Dropout Ratio | {0.2,0.} |
| | | # of Attention Heads | 8 |
| | | Attention Head Size | 64 |
| | | Attention Dropout Ratio | 0.2 |
| | Spatial Transformer | # of Transformer Layers | 4 |
| | | Hidden Size | 128 |
| | | MLP Size | 512 |
| | | MLP Dropout Ratio | {0.2,0.} |
| | | # of Attention Heads | 8 |
| | | Attention Head Size | 64 |
| | | Attention Dropout Ratio | 0.2 |
| Context CLS Head | - | Linear Projection | $256 \rightarrow 1$ |
| Optimizer | - | Batch Size | 128 |
| | | Maximum Learning Rate | 3e-4 |
| | | Minimum Learning Rate | 5e-6 |
| | | Learning Rate Scheduler | Cosine |
| | | Optimizer Type | AdamW |
| | | Adam $\beta$ | $(0.9, 0.99)$ |
| | | Weight Decay | 0.05 |
| | | Total Epochs | 100 |
| | | Warm-up Epochs | 10 |

*Table 3.* The hyperparameters for functional clustering pre-training.

| Module | Sub-Module | Name | Value |
|---|---|---|---|
| Neural Encoder | Patch Tokenizer | # of Input Channels | {1,128,128,128} |
| | | # of Output Channels | {128,128,128,128} |
| | | Kernel Size | {9,7,3,3} |
| | | Stride | {5,4,2,2} |
| | | Padding | {4,3,1,1} |
| | Prototype Embeddings | # of Prototypes | 8 |
| | Spatial Transformer | # of Transformer Layers | 4 |
| | | Hidden Size | 128 |
| | | MLP Size | 512 |
| | | MLP Dropout Ratio | {0.2,0.} |
| | | # of Attention Heads | 8 |
| | | Attention Head Size | 64 |
| | | Attention Dropout Ratio | 0.2 |
| | Temporal Transformer | # of Transformer Layers | 4 |
| | | Hidden Size | 128 |
| | | MLP Size | 512 |
| | | MLP Dropout Ratio | {0.2,0.} |
| | | # of Attention Heads | 8 |
| | | Attention Head Size | 64 |
| | | Attention Dropout Ratio | 0.2 |
| Neural Decoder | Temporal Transformer | # of Transformer Layers | 4 |
| | | Hidden Size | 128 |
| | | MLP Size | 512 |
| | | MLP Dropout Ratio | {0.2,0.} |
| | | # of Attention Heads | 8 |
| | | Attention Head Size | 64 |
| | | Attention Dropout Ratio | 0.2 |
| | Spatial Transformer | # of Transformer Layers | 2 |
| | | Hidden Size | 128 |
| | | MLP Size | 512 |
| | | MLP Dropout Ratio | {0.2,0.} |
| | | # of Attention Heads | 8 |
| | | Attention Head Size | 64 |
| | | Attention Dropout Ratio | 0.2 |
| | Time RGS Head | # of Input Channels | {128,128,128,128} |
| | | # of Output Channels | {128,128,128,128} |
| | | Kernel Size | {3,3,7,9} |
| | | Stride | {2,2,4,5} |
| | | Padding | - |
| | | Output Padding | - |
| | | Linear Projection | $128 \rightarrow 1$ |
| Optimizer | - | Batch Size | 128 |
| | | Maximum Learning Rate | 3e-4 |
| | | Minimum Learning Rate | 5e-5 |
| | | Learning Rate Scheduler | Cosine |
| | | Optimizer Type | AdamW |
| | | Adam $\beta$ | $(0.9, 0.99)$ |
| | | Weight Decay | 0.01 |
| | | Total Epochs | 100 |
| | | Warm-up Epochs | 10 |

# E. Visualization of Functional Context Prediction

The efficacy of our proposed functional context prediction (FCP) task is validated by its training dynamics. As shown in Figure 5, the contrastive loss decreases steadily while the prediction accuracy for identifying replaced channels converges rapidly to over 95%. This high and stable accuracy demonstrates that the model effectively learns to infer whether a channel is replaced from the context provided by other channels, rather than relying on intra-channel temporal correlations or trivial shortcuts. This stage successfully instills a robust understanding of inter-channel functional dependencies.

Consequently, the functional dependency graph derived from the attention patterns of this pre-trained model provides a high-quality, data-driven estimate of channel connectivity. The reliability of this graph is foundational for the second stage of our framework, as it supplies the essential supervisory signal to guide the functional clustering process. The strong performance of the downstream NeuroCLUS is therefore intrinsically linked to the success of this functional context extraction phase, which reliably captures the functional context necessary for meaningful region discovery.

We further quantify graph sparsity by directly averaging the normalized channel connectivity matrix. Comparing the graphs obtained with FCP versus masked signal prediction (MSP) yields average values of 0.28 and 0.36, respectively, providing evidence for the superiority of FCP over MSP (Figure 4 (a)).

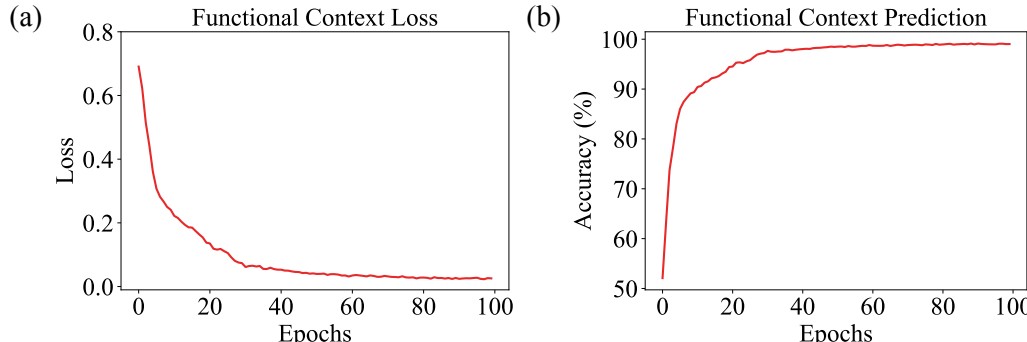

*Figure 5.* **Visualization of Functional Context Prediction. (a).** The convergence curve of functional context loss. **(b).** The convergence curve of functional context prediction.

# F. Visualization of Pre-training NeuroCLUS

The learning dynamics of pre-training NeuroCLUS are visualized in Figure 6. The overall pre-training loss converges stably to approximately 0.65, indicating effective optimization. Crucially, the reconstructed neural signals demonstrate high temporal fidelity, precisely aligning with the original signal trends. This confirms that the model successfully preserves fine-grained information while learning compressed representations.

Furthermore, the learned functional clusters align strongly with the estimated functional connectivity. The correlation coefficient between the final soft cluster assignments and the inter-channel connectivity exceeds 0.6, validating that the clustering process is meaningfully guided by the learned functional context. Together, these results demonstrate that this stage successfully balances faithful signal reconstruction with the discovery of coherent functional neural modules.

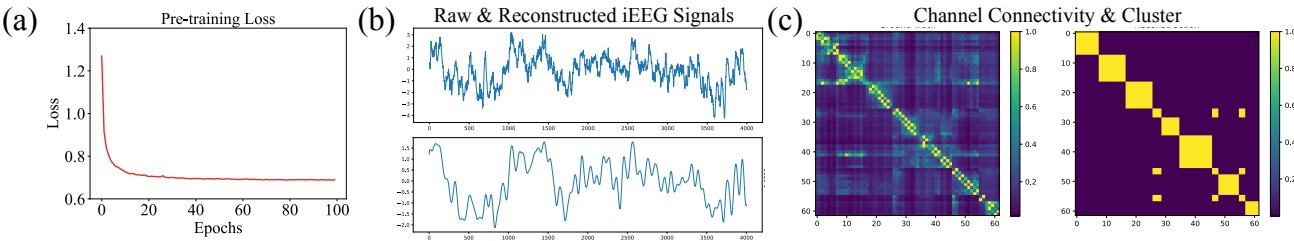

*Figure 6.* **Visualization of Pre-training NeuroCLUS. (a).** The convergence curve of pre-training loss. **(b).** The original and reconstructed iEEG signals. **(c).** The estimated channel connectivity and clusters.

## G. Visualization of Prototype Assignment

We provide a neuroscience-based interpretation of the clustering results in Figure 7. It is worth noting that, although the functional context prediction (FCP) stage extracts functional connectivity in a purely data-driven manner, the primary goal of NeuroCLUS is to use this information to group channels into distinct brain region prototypes. While our model does not explicitly encode anatomical priors, analysis on the Brain Treebank dataset reveals a clear modular structure when aggregating attention weights of prototypes across both channels and subjects. Specifically, these modules exhibit anatomical continuity – frontal electrodes tend to be assigned to one prototype, while temporal electrodes are assigned to another. We further computed the ratio of intra-cluster distance to inter-cluster distance, yielding a value of $0.1652\pm0.0251$, which quantitatively confirms the anatomical coherence of the clustering. Despite individual variability in the exact assignment of single channels, the spatial topological patterns of the prototypes remain consistent across the vast majority of subjects. Together, these findings demonstrate that NeuroCLUS can automatically learn functional modules that align with anatomical structure in a data-driven fashion, thereby achieving strong cross-subject generalization.

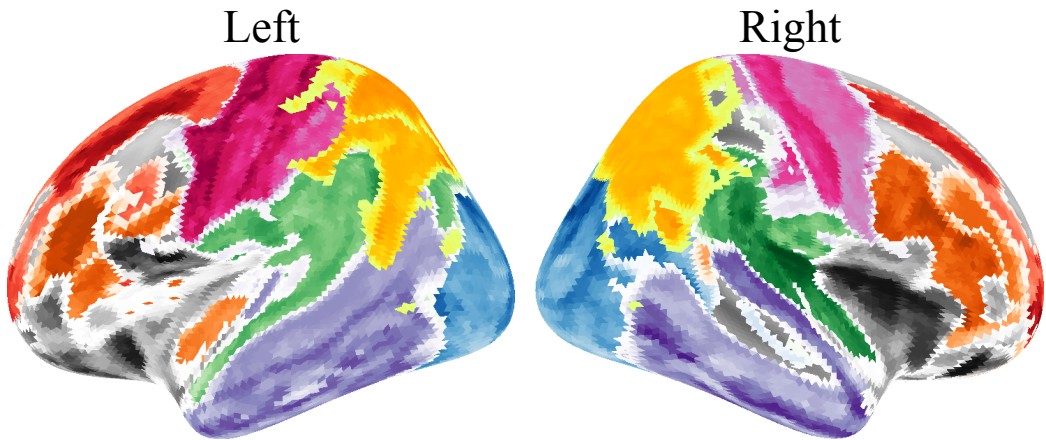

*Figure 7.* **Visualization of Prototype Assignment.**

## H. Comparison with Traditional ML Methods

Compared with traditional machine learning (ML) methods, contemporary deep learning models offer substantial advantages for modeling temporal dependencies. In the context of spontaneous neural activity – exemplified by the Du-IN speech production task (Zheng et al., 2025) – neural signals lack explicit temporal anchors. As a result, traditional machine learning methods struggle to capture the underlying dynamic temporal structure, making the strengths of deep learning particularly evident (SVM: 7.82%; GLM: 5.49%). In contrast, for event-evoked neural activity, such as that found in the Brain Treebank benchmark (Wang et al., 2024a), characteristic waveforms are already aligned relative to stimulus onset. Under these conditions, the performance gap between traditional machine learning and deep learning narrows considerably. Nonetheless, the original BrainBERT (Wang et al., 2023) experiments still demonstrate clear advantages of deep learning approaches. Building on this foundation, NeuroCLUS further incorporates functional clustering, achieving state-of-the-art results across both task categories, as reported in Table 4.

*Table 4.* Comparison with traditional ML methods across 5 iEEG datasets and 6 tasks. The best results are **bolded**, while the second results are underlined.

| Methods | SWEC | | MAYO | FNUSA | Brain Treebank | | | | Du-IN |
|---|---|---|---|---|---|---|---|---|---|
| | Seizure | | Seizure | Seizure | Pitch | Volumn | Onset | Speech | Word |
| | Kappa | F1 | F1 | F1 | ROC-AUC | ROC-AUC | ROC-AUC | ROC-AUC | BAC (%) |
| GLM | 0.12±0.03 | 0.09±0.04 | 0.04±0.03 | 0.08±0.05 | 0.58±0.07 | 0.56±0.19 | 0.63±0.04 | 0.58±0.06 | 5.49±0.52 |
| SVM | 0.17±0.04 | 0.12±0.04 | 0.07±0.04 | 0.10±0.04 | 0.60±0.06 | 0.57±0.08 | 0.67±0.03 | 0.62±0.04 | 7.82±0.86 |
| NeuroCLUS | **0.61±0.01** | **0.59±0.02** | **0.40±0.02** | **0.51±0.02** | **0.83±0.02** | **0.92±0.02** | **0.96±0.01** | **0.99±0.01** | **65.92±4.17** |

## I. Ablations on Functional Context Prediction

In addition to the results shown in Figure 4 (a), Table 5 further reports model performance when guided by coherence-based similarity (COH). The observed performance gap indicates that low-level coherence alone is insufficient to fully capture the functional relationships among channels, leading to poor performance.

*Table 5.* Ablations to explore the impact of functional context prediction across 5 iEEG datasets and 6 tasks. The best results are **bolded**, while the second results are underlined.

| Methods | SWEC | | MAYO | FNUSA | Brain Treebank | | | | Du-IN |
|---|---|---|---|---|---|---|---|---|---|
| | Seizure | | Seizure | Seizure | Pitch | Volumn | Onset | Speech | Word |
| | Kappa | F1 | F1 | F1 | ROC-AUC | ROC-AUC | ROC-AUC | ROC-AUC | BAC (%) |
| COH | 0.59±0.01 | 0.56±0.03 | 0.39±0.02 | 0.49±0.03 | 0.78±0.03 | 0.89±0.03 | 0.93±0.02 | 0.96±0.02 | 57.20±4.82 |
| MSP | 0.61±0.01 | 0.58±0.02 | 0.40±0.02 | 0.50±0.02 | 0.81±0.02 | 0.91±0.03 | 0.94±0.02 | 0.98±0.01 | 61.34±4.92 |
| FCP | **0.61±0.01** | **0.59±0.02** | **0.40±0.02** | **0.51±0.02** | **0.83±0.02** | **0.92±0.02** | **0.96±0.01** | **0.99±0.01** | **65.92±4.17** |

## J. Ablations on Pre-training Loss

Figure 4 (c) presents the ablation results on the pre-training loss components. Detailed explanations of the observed performance on the Du-IN speech production task are as follows:

**NeuroCLUS w/o prototypes:**   The NeuroCLUS variant that completely removes the prototype embedding mechanism, degrading the model to a purely channel-level architecture, akin to Brant (Zhang et al., 2023) or BaRISTA (Oganesian et al., 2025). In this configuration, performance drops to 16.90%, placing it on par with channel-level baselines.

**NeuroCLUS w/o $\mathcal{L}_{clus}$:**   The NeuroCLUS variant pre-trained with only the reconstruction loss $\mathcal{L}_{mse}$. While it retains the prototype embedding mechanism to enforce channel-to-region aggregation, it omits $\mathcal{L}_{clus}$ and therefore does not leverage the functional connectivity graphs extracted by the functional connectivity extractor. This model achieves 44.72% balanced accuracy, demonstrating that the prototype embedding mechanism alone already yields substantially better performance than purely channel-level architectures (e.g., Brant (Zhang et al., 2023), MVPFormer (Carzaniga et al., 2025), and etc.).

**NeuroCLUS w/o $\mathcal{L}_{mse}$:**   The NeuroCLUS variant pre-trained with only the clustering loss $\mathcal{L}_{clus}$. In this setting, the latent space collapses and fails to capture the underlying structure of signal distribution, leading to a performance drop to 32.77%.

**NeuroCLUS:**   The full NeuroCLUS model pre-trained with both $\mathcal{L}_{mse}$ and $\mathcal{L}_{clus}$. Compared to NeuroCLUS w/o $\mathcal{L}_{clus}$ (44.72%), the full NeuroCLUS achieves 65.92%, highlighting the pivotal role played by $\mathcal{L}_{clus}$ – which utilizes the functional connectivity graphs extracted by the functional connectivity extractor – in optimizing the prototypes.

Therefore, the performance improvements of NeuroCLUS over the channel-level baselines (16.90% (NeuroCLUS w/o prototypes) → 44.72% (NeuroCLUS w/o $\mathcal{L}_{clus}$) → 65.92% (NeuroCLUS)) arise from the combined contributions of the prototype embedding mechanism and the functional clustering modeling.

## K. Ablations on Pre-training Loss Weight

In practice, we set the total pre-training loss as $\mathcal{L} = \mathcal{L}_{mse} + \mathcal{L}_{clus}$ without additional weighting, since both loss terms naturally exhibit similar numerical scales. To further quantify the impact of the relative weights between the two objectives, we reformulate the loss as $\mathcal{L} = \lambda_{mse} \cdot \mathcal{L}_{mse} + \lambda_{clus} \cdot \mathcal{L}_{clus}$, where $\lambda_{mse}$ and $\lambda_{clus}$ are adjustable coefficients. We then perform an ablation study over the ratio $\frac{\lambda_{clus}}{\lambda_{mse}}$, with results reported in Table 6. The findings show that performance remains stable across a range of ratios, with fluctuations of only $\sim 2\%$. This robustness indicates that the pre-training objectives are well-balanced and that the model does not rely on fine-tuned weighting for effective learning.

*Table 6.* Ablations to explore the impact of the pre-training loss weight across 5 iEEG datasets and 6 tasks. The best results are **bolded**, while the second results are underlined.

| $\frac{\lambda_{clus}}{\lambda_{mse}}$ | SWEC | | MAYO | FNUSA | Brain Treebank | | | | Du-IN |
|---|---|---|---|---|---|---|---|---|---|
| | Seizure | | Seizure | Seizure | Pitch | Volumn | Onset | Speech | Word |
| | Kappa | F1 | F1 | F1 | ROC-AUC | ROC-AUC | ROC-AUC | ROC-AUC | BAC (%) |
| 0.2 | 0.60±0.01 | 0.58±0.01 | 0.38±0.01 | 0.50±0.01 | 0.82±0.02 | 0.92±0.02 | 0.96±0.01 | 0.99±0.01 | 65.18±4.02 |
| 0.5 | 0.61±0.02 | 0.58±0.01 | 0.39±0.01 | 0.51±0.02 | 0.83±0.03 | 0.91±0.02 | 0.96±0.01 | 0.99±0.01 | 65.51±3.96 |
| 1.0 | **0.61±0.01** | 0.59±0.02 | **0.40±0.02** | **0.51±0.02** | 0.83±0.02 | **0.92±0.02** | **0.96±0.01** | **0.99±0.01** | **65.92±4.17** |
| 2.0 | 0.60±0.01 | **0.59±0.02** | 0.40±0.02 | 0.51±0.02 | **0.84±0.03** | 0.91±0.02 | 0.96±0.01 | 0.99±0.01 | 65.59±4.28 |
| 5.0 | 0.60±0.02 | 0.59±0.01 | 0.39±0.01 | 0.50±0.02 | 0.83±0.03 | 0.91±0.02 | 0.95±0.01 | 0.98±0.01 | 65.60±3.91 |

## L. Model Efficiency

For a fixed input duration, we estimated the MFLOPs of each model using the `thop` package with results reported in Table 7. All measurements were obtained on a single NVIDIA 3090 GPU. The computational cost of NeuroCLUS (834.82 MFLOPs) is comparable to that of PopT (Chau et al., 2024) and BaRISTA (Oganesian et al., 2025), and substantially lower than that of Brant (Zhang et al., 2023) and MVPFormer (Carzaniga et al., 2025). These results indicate that NeuroCLUS achieves state-of-the-art performance while maintaining a relatively lightweight computational footprint, making it well-suited for deployment as a foundation model in practical applications.

*Table 7.* Model Efficiency Analysis on NeuroCLUS and baselines.

| Methods | Brant | LaBraM | Du-IN | H2DiLR | PopT | BaRISTA | MVPFormer | NeuroCLUS |
|---|---|---|---|---|---|---|---|---|
| MFLOPs | 1522.98 | 354.48 | **159.69** | 539.07 | 788.94 | 642.18 | 2194.69 | 834.82 |

## M. Zero-shot Evaluation

Table 8 reports the zero-shot evaluation results of NeuroCLUS on unseen subjects. For the epilepsy monitoring tasks (i.e., SWEC, MAYO, and FNUSA), we adhere strictly to the evaluation protocol established by MVPFormer (Carzaniga et al., 2025), which is inherently a cross-subject zero-shot setting. The results obtained are consistent with those shown in Table 1. For higher-order cognitive tasks (i.e., Brain Treebank and Du-IN), zero-shot performance exhibits a substantial drop compared to the fine-tuned setting. This gap indicates that decoding complex cognitive processes is considerably more challenging under zero-shot conditions, underscoring the need for task-specific fine-tuning. All evaluations on Brain Treebank and Du-IN employ a leave-one-subject-out zero-shot scheme. Specifically, on the Brain Treebank benchmark, we train NeuroCLUS on six subjects and evaluate on the held-out subject.

*Table 8.* Zero-shot evalution of NeuroCLUS across 5 iEEG datasets and 6 tasks.

| - | SWEC | | MAYO | FNUSA | Brain Treebank | | | | Du-IN |
|---|---|---|---|---|---|---|---|---|---|
| | Seizure | | Seizure | Seizure | Pitch | Volumn | Onset | Speech | Word |
| | Kappa | F1 | F1 | F1 | ROC-AUC | ROC-AUC | ROC-AUC | ROC-AUC | BAC (%) |
| NeuroCLUS | 0.61±0.01 | 0.59±0.02 | 0.40±0.02 | 0.51±0.02 | 0.54±0.01 | 0.57±0.02 | 0.65±0.02 | 0.62±0.02 | 6.92±0.80 |

