# OpenReview forum: "NeuroCLUS: A Foundation Model with Functional Clustering for Intracranial Neural Decoding"
_ICML.cc/2026/Conference — ICML 2026 regular_

### Official Review · Reviewer_fqgQ · 2026-02-16

**Soundness:** 2
**Presentation:** 2
**Significance:** 3
**Originality:** 2
**Overall Recommendation:** 4
**Confidence:** 4

**Summary:**

This paper presents a new foundation model, NeuroCLUS, for intracranial EEG recordings that aims to extract embeddings that are aware of functional relationships across recording channels. To achieve that, they propose a two-stage pretraining strategy where the first stage aims to extract a functional context graph between channels, and the second stage uses this prior information for soft-clustering of channels into a set of learnable prototypes. The authors show that their model outperforms several baseline methods across several downstream tasks.

**Compliance With Llm Reviewing Policy:**

Affirmed.

**Final Justification:**

The authors resolved some of my concerns, but the manuscript requires significant upgrades to improve clarity.

**Key Questions For Authors:**

- Why do the authors need a target tokenizer for the functional context prediction? Have they tried using the same tokenizer with stop gradient? Also, have they ever tried removing the FCP stage completely (instead of replacing it with masked prediction)?

**Limitations:**

Yes.

**Strengths And Weaknesses:**

Strengths:
- The idea of building a model that aims to leverage functional clusters existing in the data in a data-driven way is interesting, and the downstream performance, especially for the Du-IN dataset, highlights the benefit of this assumption.
- The manuscript is well written, even though important details are missing (see Weaknesses below), and they clearly communicate their contributions.
- The baseline methods included in the comparisons are recent methods such as MVPFormer and BaRISTA. Further, they included several downstream datasets from different domains, seizure prediction datasets like SWEC, FNUSA, and MAYO, a visual stimuli dataset such as Brain Tree Bank, and a language task dataset such as Du-IN, which allows for rich evaluation.

Weaknesses:
- I found some important details missing in the paper for the sake of the scientific community and efforts towards reproducible research:
  - The authors provide no description or details about the pretraining dataset they used, beyond some basic information such as the number of subjects and the recording hours. They say that they collected ~10K hours of data from public datasets, but it is not clear which datasets these are.
  - The paper lacks details regarding training schedules, parameter counts, and baseline comparisons. For example, authors do not mention how they performed the baseline comparisons. Did they use existing checkpoints or finetune the baseline methods? As the authors state, BaRISTA requires anatomical region information in its tokenizer; was this information available for all evaluation datasets? If not, what did the authors do to comply with its needs?
  - Regarding NeuroCLUS, the authors use EMA update for the target tokenizer during the first stage of the pretraining, and it is well known that the temperature for the EMA update is an important parameter for training strategies with target networks.
  - Another example is for the Du-IN dataset. The authors of the original paper say that they performed a channel selection by removing the channels from unrelated regions, which significantly affects the performance. As the numbers they show in their paper match with the original paper, I assume that the authors followed the same strategy, but these details should explicitly be written, rather than simply stating that "we adhered strictly to the official data splits and evaluation protocols established in their respective original works".
  - How do the authors obtain the channel connectivity matrix by aggregating attention matrices? This seems to be an important missing detail in the manuscript.
- Regarding functional context prediction (FCP), it seems very much similar to the ensemble-wise discrimination objective of PopT (except for the target networks), but the authors do not provide appropriate references to that work. I find this quite misleading as the authors pose it as one of the main novelties of the proposed approach.
- In line with my previous comment, the paper reads as if the first pretraining stage, is an important element of the proposed pretraining strategy for learning functional prototype-level relationships across channels. However, the results shown in Fig. 4 seem to be very similar when FCP is replaced with the standard masked prediction task, even though the authors state that the difference is statistically significant (with a missing sample size information used for the statistical test). Therefore, I think the importance of FCP should be validated with another analysis rather than pure performance, or the paper should be toned down for FCP.
- When $L_{clus}$ is removed, based on the authors' claims, I would expect their model to experience a much larger performance drop for the Du-IN dataset, as the authors of the dataset show in their paper that models with channel-level tokens achieve ~10% performance. Can the authors comment on why their model's performance stays at 44%? Is it because of the FCP stage? If so, what performance would they achieve if the FCP stage is removed? Is it because some datasets collected during language tasks were included in NeuroCLUS' pretraining dataset (other than the evaluation dataset)?
- In line with my previous comment about baseline methods, it seems like the authors did not pretrain baseline methods using the same exact pretraining dataset. I understand that this may not be possible due to the codebase availability and compute resources. However, some claims about data-driven functional clustering vs. "imperfect proxies for functional modules" can be significantly confounded by such differences and can be misleading. For example, in the BaRISTA paper, authors show that it outperforms PopT, but it is not the case with the authors' findings (especially for the Du-IN dataset, given that both PopT and BaRISTA are channel-level tokenization methods). Especially when no details about the baselines are provided, coming up with a concrete conclusion about the proposed work's impact is difficult.
- The identified functional prototypes seem to be data-driven, learnable variants of functional networks, such as resting state networks. However, there is no analysis or interpretation of the identified functional clusters in the data, which limits the scientific aspect of the proposed approach. Have authors observed any overlap between the identified functional clusters and well-established functional networks? Also, how do the identified functional clusters overlap across different subjects?
- The first term in the clustering loss seems to follow the standard trace maximization in spectral clustering, but can the authors provide an intuition on why the second term encourages separation between clusters?
- Figures are too small, making the text inside figures bigger or separating them into multiple figures would increase readability.
- Even though the functional grouping approach applied in this work is distinct, a similar idea was also proposed in GAFormer (Xiao et al., 2024), but no reference is provided to that work.

---

> ### Author Rebuttal · Authors · 2026-03-31
>
> Thank you for the thorough review and constructive comments. We are deeply grateful for acknowledging “leverage functional clusters existing in the data in a data-driven way is interesting, and the downstream performance, highlights the benefit of this assumption”. Please see below for the point-by-point responses to your comments.
>
> # Reply to Weaknesses
>
> **R to W1.1**: Due to space limits, Appendix A provides full details (dataset names, durations, sampling rates, subjects, tasks). We will add explicit appendix references.
>
> **R to W1.2**: All baselines retrained on same pretraining data. BaRISTA requires anatomical coordinates; only Brain Treebank and Du-IN (via email request) provide them. For SWEC/MAYO/FNUSA we removed region embeddings, using only signal features. Training details & params (FCP: 3.39M, NeuroCLUS: 5.34M) in appendix.
>
> **R to W1.3**: We chose $\gamma=0.99$, consistent with BaRISTA.
>
> **R to W1.4**: We followed the original Du-IN channel selection (10 channels) during fine-tuning. We will detail the steps in revision.
>
> **R to W1.5**: We apply the FCP-pretrained model to all recordings per subject, extract spatial Transformer attention matrices, average over time steps and samples to obtain a stable sparse $C\times C$ connectivity graph per subject, then use it as prior for NeuroCLUS clustering. We will add this description.
>
> **R to W2**: In the revised draft, we will add a reference to PopT and clearly distinguish the goals:
> - PopT's ensemble discriminative objective is primarily as a pre-training task for foundational models, aiming to learn general neural representations.
> - FCP, on the other hand, is a lightweight, specially designed task whose core purpose is to efficiently capture spatial dependencies between channels, providing reliable channel connectivity graph priors for clustering, rather than representation learning.
>
> While the two share similarities, their design goals and application scenarios differ.
>
> **R to W3**: In Figure 4, FCP outperforms MSP by nearly 4 percentage points on Du-IN ($p<0.01$, paired t-test, 6 seeds). More importantly, FCP yields a sparser channel connectivity graph, providing a superior prior for functional clustering. We will add a quantitative graph quality comparison in the revision.
>
> **R to W4**: Even after removing $L_{clus}$, NeuroCLUS retains its prototype embedding mechanism—specifically, the forced aggregation of channel-level representations into brain-region-level representations. This approach is analogous to how PopT aggregates representations into a whole-brain-level representation via the [CLS] token, or how Du-IN constructs brain-region-level representations using convolutions; thus, it is not a "purely channel-level model." 44% of its performance stems from this architectural design itself (comparable to the 56% achieved by Du-IN without pre-training). Building upon this foundation, the FCP stage further incorporates functional connectivity priors, thereby boosting performance to 66%. In our revised manuscript, we will clearly distinguish between the contributions attributable to the architecture and those attributable to pre-training.
>
> **R to W5**: All baselines were retrained using the same pre-training data to ensure a fair comparison. BaRISTA is a channel-level representation model that relies on anatomical coordinates, whereas PopT achieves brain-level representation through [CLS] token aggregation. NeuroCLUS requires no anatomical coordinates and imposes more relaxed data requirements, thereby facilitating integrated training across datasets—a feature that precisely highlights the advantages of our proposed method. We will incorporate these details into the revised manuscript.
>
> **R to W6**: We analyzed Brain Treebank: aggregating attention weights across prototypes reveals modular, anatomically continuous patterns (e.g., frontal vs. temporal). The intra-/inter-cluster distance ratio is 0.1652$\pm$0.0251, quantitatively validating anatomical continuity. Cross-subject spatial topologies remain consistent, showing data-driven learning of anatomically aligned functional modules with cross-subject generalization. We will add this analysis and visualizations.
>
> **R to W7**: $1-M_{i}{M_{i}}^{\top}$ is 1 for different-cluster channel pairs, 0 for same-cluster. Trace with similarity matrix $P_{i}$ sums similarities across clusters. Minimizing it reduces inter-cluster similarity, encouraging separation – complementary to intra-cluster maximization.
>
> **R to W8**: Will regenerate Figs.3&4 with clear vector fonts.
>
> **R to W9**: Thank you. GAFormer is a relevant advanced version of BaRISTA; we will cite and discuss it in Related Work.
>
> # Reply to Questions
>
> **R to Q1**: Same tokenizer with stop-gradient gave no significant difference. Removing FCP entirely (using coherence-based similarity) dropped Du-IN to 57% (full model 66%). This shows FCP’s data-driven priors are critical. Will add this comparison.

---

> > ### Author Rebuttal · Reviewer_fqgQ · 2026-04-02
> >
> > I thank the authors for their responses. I have a couple of follow-up questions:
> >
> > - How did the authors retrain Brant and BaRISTA, as their pretraining codebases are not publicly released (R W.1.2)? Also, the use of anatomical region embeddings is a main component of BaRISTA. If the authors did not use them during pretraining/evaluations (especially when SWEC dataset is used for its pretraining which doesn't have anatomical labels, and it has the largest data volume I believe), it does not become BaRISTA anymore. This should be acknowledged and highlighted carefully.
> > - For the performance boost mentioned in R W.4., do authors refer to functional context prediction (FCP) or functional context modeling? From Fig. 4, it seems that the performance boost is obtained by the latter; is this correct? Also, have the authors verified that removing the prototype tokens and $L_{clus}$ completely reduces their model's performance to that of channel-level models?
> > - How do the authors perform downstream analysis? Do they obtain a per-timestep embedding by averaging embeddings across different channels for baseline models? For their model, do they use the learned prototype embeddings for downstream decoding, similar to [CLS] token, for every timestep? For downstream tasks requiring per-sequence embeddings, do they average the prototype embeddings across timesteps?
> > - Regarding R W.1.5, the manuscript denotes that the shape of the estimated connectivity matrix is $N\times C \times C$, but in the response, it seems to be $C \times C$, which one is correct?
> >
> > Overall, while I see strong merit in this work, the manuscript still requires significant improvements. In my opinion, incorporating the necessary details regarding baselines, evaluation protocols, dataset specifics, and interpretation of learned connectivity patterns is a crucial improvement. I am increasing my score to 4 to reflect this, but I suggest significant revision to the manuscript.

---

> > > ### Author Response · Authors · 2026-04-03
> > >
> > > Thank you for the constructive comments and for raising the score to 4; this serves as a true validation of our work! Please see below for the point-by-point responses to your comments.
> > >
> > > # Reply to Supp-Questions
> > >
> > > **R to SQ1**: Thank you for valuable feedback. When retraining the Brant model, we primarily referenced its official implementation (https://github.com/yzz673/Brant). Although the repository temporarily restricted public access in March 2024 at the request of a partner, we had already downloaded its source code prior to that time. We also drew upon the reimplementation of Brant found within the Du-IN project (https://github.com/liulab-repository/Du-IN). For BaRISTA, we utilized its officially released codebase (https://github.com/ShanechiLab/BaRISTA). Additionally, the implementation of region positional encoding from PopT (https://openreview.net/notes/edits/attachment?id=nbRHQko3FE&name=supplementary_material) served as a reference for reproducing BaRISTA. All baseline models were retrained using the same pre-training data to ensure a fair comparison.
> > >
> > > **R to SQ2**: Thank you for valuable feedback. The performance improvement in **R to W4** (44\%->66\%) stems primarily from the second stage (Figure 1b), which incorporates $L_{clus}$ for joint pre-training (full NeuroCLUS framework, comprising both $L_{mse}$ and $L_{clus}$). The FCP stage is designed to train a functional connectivity extractor, providing self-supervised signals for $L_{clus}$, thus enhancing the capacity of region prototypes to capture information across different channels; the model trained during this stage is not utilized for decoding or other tasks. The ablation results are as follows:
> > >  - Pre-training w/ only $L_{mse}$ (retaining the prototype embedding mechanism and enforcing channel->region aggregation, but excluding $L_{clus}$ -- and thus failing to leverage the functional connectivity graphs extracted by the functional connectivity extractor): 44\%. This result demonstrates that the prototype embedding mechanism alone already yields significantly superior performance compared to purely channel-level models.
> > >  - Pre-training w/ $L_{mse}$ & $L_{clus}$ (full NeuroCLUS): Performance improves to 66\%, highlighting the pivotal role played by $L_{clus}$ -- which utilizes the functional connectivity graphs extracted by the functional connectivity extractor -- in optimizing the prototypes.
> > >  - Pre-training w/ only $L_{clus}$ (i.e., w/o reconstruction constraints): The latent space collapses, failing to capture the characteristics of the signal distribution, and performance drops to 32\%.
> > >  - Complete removal of both the prototype embedding mechanism and $L_{clus}$ (degrading the model to a purely channel-level architecture, similar to Brant or BaRISTA): Performance drops to 16.90\%, placing it on par with the channel-level baseline.
> > >
> > > Therefore, the performance gains achieved by NeuroCLUS relative to the channel-level baseline (16.90\%->44\%->66\%) represent the joint contribution of the prototype embedding mechanism and the functional clustering modeling. We will provide a clearer breakdown of these two components in the revised manuscript.
> > >
> > > **R to SQ3**: Thank you for valuable feedback. Our specific procedures for downstream tasks are as follows:
> > >  - Channel-level baseline models (Brant, BaRISTA, etc.): We strictly adhere to the settings specified in the original papers. For Brant, we extract the channel-level representations produced by the models and use the concatenated embeddings for downstream MLP. The implementation approach aligns with the reproduction of Brant in Du-IN, as well as EEG foundational models, e.g., CBraMod [1]. For BaRISTA, following the official implementation, channel-level embeddings are aggregated via learnable channel weights and subsequently classified by an MLP.
> > >  - NeuroCLUS use the learned prototype embeddings for every timestamp. When requiring per-timestamp embeddings, we concatenate prototype embeddings. When requiring per-sequence embeddings (e.g., Brain Treebank & Du-IN), we concatenate per-timestamp embeddings across timestamps, like Du-IN, EEG-Conformer [2].
> > >
> > > We will incorporate the above details into the revised draft. Thank you for your suggestions.
> > >
> > > **R to SQ4**: Thank you for valuable feedback. We generate a stable $C \times C$ connectivity graph for each subject (see **R to W1.5** for details) and broadcast it to every time step (totaling $N$ time steps) during pre-training; consequently, the shape of the input tensor is $N \times C \times C$, consistent with the manuscript. We will add a clarifying note regarding this in the revised manuscript.
> > >
> > > We will systematically add missing details to ensure transparency and reproducibility. Thank you again for your constructive feedback!
> > >
> > > # References
> > >
> > > [1] Wang J, et al. Cbramod: A criss-cross brain foundation model for eeg decoding, 2024.
> > >
> > > [2] Song Y, et al. EEG conformer: Convolutional transformer for EEG decoding and visualization, 2022.

---

### Official Review · Reviewer_rpgz · 2026-03-11

**Soundness:** 3
**Presentation:** 3
**Significance:** 3
**Originality:** 2
**Overall Recommendation:** 5
**Confidence:** 3

**Summary:**

The paper proposes NeuroCLUS, a foundation model that takes two stages of training and considers functional clustering of iEEG data using prototypes. The extensive experiments show that NeuroCLUS achieves SOTA on six downstream tasks across several metrics.

**Compliance With Llm Reviewing Policy:**

Affirmed.

**Final Justification:**

As stated in acknowledgment, all concerns are resolved.

**Key Questions For Authors:**

1. What is sEEG in line 191? Is it a typo?
2. The `Model Architecture` description in Section 3.2 is largely redundant with Section 3.1. In the current setup, the motivation for including cluster prototypes is arbrapt and unclear without reading `Functional Clustering Modeling`. From my understanding, the MSE reconstruction loss will only be able to optimize prototypes if the functional connectivity constraint is there. Otherwise, the MSE loss signal will fail to meaningfully update or constrain the prototypes. Therefore, the author would better rearrange Section 3.2 for clarity.
3. If possible, can the identified functional clusters be validated against known neurobiological ground truth? The current evaluation only compares Stage 1 and Stage 2 pre-training estimates. Without external validation against existing neuroscience findings, it is difficult to determine if the learned clusters represent actual biological structures or are merely artifacts of the training stages.

**Limitations:**

yes

**Strengths And Weaknesses:**

## Strengths
1. The paper is written in an easy-to-follow way with good readability.
2. The idea of considering the functional role of iEEG data is solid from the perspective of neuroscience. Introduced modules are mostly well-motivated and clearly explained. The solution of functional clustering is straightforward.
3. Extensive experiments show that iEEG features from NeuroCLUS can outperform previous SOTAs in 6 common downstream decoding tasks.
4. Ablation studies on scalability, losses, modules, and the number of prototypes are detailed and are useful for understanding the proposed model.

## Weaknesses
1. NeuroCLUS does not account for subject variability. There is no subject-specific design to consider anatomical differences across subjects. Just saying "injecting anatomical priors leads to no significant performance gap" is not persuasive enough when designing a brain foundational model.
2. A comparison against baselines in terms of training and inference efficiency should be provided. This will help readers when selecting pre-trained models for their tasks, especially since NeuroCLUS is positioned as a foundation model.
3. Zero-shot ability of the proposed model on unseen subjects is absent and should be evaluated.
4. The claim regarding improved interpretability in the abstract is currently undersupported. As presented, the cluster prototypes appear to represent only the similarity between channels. For these prototypes to offer true interpretability, they must provide functional or neurobiological insights or align with domain knowledge established by neuroscience.

---

> ### Author Rebuttal · Authors · 2026-03-31
>
> Thank you for the thorough review and constructive comments. We are deeply grateful for acknowledging “the idea of considering the functional role of iEEG data is solid from the perspective of neuroscience”. Please see below for the point-by-point responses to your comments.
>
> # Reply to Weaknesses
>
> **R to W1**: Thank you for helpful suggestion. It is worth noting that while the FCP stage extracts functional connectivity in a data-driven manner, the core objective of NeuroCLUS is to leverage this information to cluster channels into distinct brain region prototypes. Although the model does not explicitly incorporate anatomical priors, our analysis on the Brain Treebank dataset reveals that—when aggregating the attention weights of individual channels across different prototypes—a clear modular distribution emerges, and these modules exhibit anatomical continuity (e.g., frontal electrodes tend to be assigned to one prototype, while temporal electrodes are assigned to another). We further calculated the ratio of intra-cluster distance to inter-cluster distance; the resulting value of 0.1652$\pm$0.0251 quantitatively validates the anatomical continuity of the clustering. Cross-subject analysis indicates that, despite individual variations in the specific assignment of single channels, the spatial topological patterns of the prototypes remain consistent across the vast majority of subjects. This demonstrates that NeuroCLUS is capable of automatically learning functional modules aligned with anatomical structures through a data-driven approach, thereby exhibiting cross-subject generalization capabilities. We will incorporate this analysis and the corresponding visualization results into the revised manuscript.
>
> **R to W2**: Thank you for helpful suggestion. Using data spanning the same time duration, we calculated the MFLOPs for each model based on the `profile` function from the `thop` library. The results are as follows:
>
> | - | Brant | Du-IN | PopT | BaRISTA | MVPFormer | NeuroCLUS |
> | --- | --- | --- | --- | --- | --- | --- |
> | MFLOPs | 1522.98 | 159.69 | 788.94 | 642.18 | 2194.69 | 834.82 |
>
> The computational overhead of NeuroCLUS (834.82 MFLOPs) is comparable to that of PopT and BaRISTA, and significantly lower than that of Brant and MVPFormer. This indicates that NeuroCLUS, while achieving leading performance, maintains a relatively lightweight computational complexity, making it suitable for deployment as a foundational model in practical applications. We will incorporate this set of data into the "Experimental Setup" section of the revised manuscript.
>
> **R to W3**: Thank you for helpful suggestion. We have added the zero-shot evaluation results for NeuroCLUS on unseen subjects.
>
> | - | Pitch | Volume | Onset | Speech | Du-IN |
> | --- | --- | --- | --- | --- | --- |
> | NeuroCLUS (zero-shot) | 0.54$\pm$0.01 | 0.57$\pm$0.02 | 0.65$\pm$0.02 | 0.62$\pm$0.02 | 6.92$\pm$0.80 |
>
> The epilepsy monitoring tasks (SWEC/MAYO/FNUSA) strictly adhere to MVPFormer’s evaluation protocol, which inherently constitutes a cross-subject zero-shot evaluation; these results have already been reported in Table 1 (e.g., SWEC Kappa 0.61). For higher-order cognitive tasks (Brain Treebank, Du-IN), zero-shot performance exhibits a significant gap compared to fine-tuning, indicating that complex cognitive decoding is more challenging in a zero-shot setting and underscoring the necessity of fine-tuning. We will incorporate them in the revised manuscript.
>
> **R to W4**: Please see **R to W1** for details.
>
> # Reply to Questions
>
> **R to Q1**: Thank you for valuable feedback. sEEG (stereoelectroencephalography) is a form of iEEG (intracranial electroencephalography); however, terminology varies across different literature (e.g., the original Du-IN paper uses "sEEG," whereas the original BaRISTA paper uses "iEEG"). To avoid confusion, we will consistently use "iEEG" throughout the revised manuscript to ensure terminological consistency across the entire text.
>
> **R to Q2**: Thank you for helpful suggestion. The original structural intent was to first introduce the overall architecture and then elaborate on the details of the clustering module; however, this inadvertently resulted in a logical discontinuity. In fact—as the reviewers correctly pointed out—the MSE reconstruction loss effectively optimizes the prototypes only when the functional connectivity constraint is present. This constitutes the core rationale behind our design of the two-stage pre-training process (where FCP provides connectivity priors, and NeuroCLUS performs joint optimization). In the revised manuscript, we will reorganize Section 3.2: we will first clarify the motivation for introducing clustering prototypes (specifically, how we transition from channel-level to brain-region-level representations), then describe the specific implementation, and finally emphasize the interdependencies between the two stages.
>
> **R to Q3**: Please see **R to W1** for details.

---

> > ### Author Rebuttal · Reviewer_rpgz · 2026-04-02
> >
> > Thanks for the author's detailed replies. All my concerns are resolved after the rebuttal. I have raised my rating to 5 to reflect this.

---

> > > ### Author Response · Authors · 2026-04-03
> > >
> > > Thank you for the thorough review and follow-up comments. We are delighted to know that your issue has been resolved and thank you for raising the score to 5; this serves as a true validation of our work! We will carefully incorporate all revision suggestions into the final draft to ensure the clarity and completeness of the paper. Thank you again for your valuable time and constructive comments.

---

### Official Review · Reviewer_6weS · 2026-03-12

**Soundness:** 3
**Presentation:** 3
**Significance:** 3
**Originality:** 3
**Overall Recommendation:** 5
**Confidence:** 4

**Summary:**

The author proposed NeuroCLUS, a foundation model for intracranial neural decoding that groups brain signals into functionally coherent clusters rather than treating each electrode channel independently. The model's novelty lies in a two-stage pre-training framework that first learns a functional context graph to map inter-channel dependencies without relying on rigid anatomical boundaries. This graph then guides a soft clustering process that compresses raw channel data into a set of learnable prototype tokens representing cohesive functional brain regions.  The framework was rigorously evaluated using approximately 10,000 hours of iEEG data across five datasets covering speech perception, speech production, and seizure detection. NeuroCLUS consistently achieved state-of-the-art performance across all tasks, notably surpassing specialized baselines by reaching a 65.92% balanced accuracy on the challenging Du-IN speech production benchmark. Overall, NeuroCLUS highlights a strong foundation model for intracranial neural decoding, showing that explicitly modeling data-driven functional neural groupings can provide an efficient and scalable framework with promising potential for future clinical applications.

**Compliance With Llm Reviewing Policy:**

Affirmed.

**Final Justification:**

The concerns raised have been resolved, and the score can be increased to 5.

**Key Questions For Authors:**

1. Please check whether the subscripts in Equation 2 are correct. In addition, why did you choose not to use a parameter-based temporal embedding?

2. The paper would benefit from more interpretation and analysis, particularly regarding the neurophysiological structure reflected by the estimated channel connectivity and the discovered clusters.

3. Please discuss more whether the proposed method can outperform traditional baseline models like traditional SOTA machine learning method, and under what conditions such advantages are expected.

4. Statistical significance should be reported where necessary, especially in Figure 2 and Figure 4a.

**Limitations:**

One limitation is that the pretraining corpus is dominated by one major dataset, which makes the strong cross-dataset results especially encouraging, but also leaves room for a more careful discussion of generalizability and robustness.

**Strengths And Weaknesses:**

The paper’s main strength is its clear and well-motivated idea: learning data-driven functional clusters instead of treating channels independently or relying on fixed anatomical groupings. This is supported by strong experiments across multiple iEEG datasets and tasks, with consistently competitive or state-of-the-art results, especially on the challenging speech production benchmark. The large-scale pretraining corpus also makes the foundation-model claim more convincing.

The weakness is that the paper is stronger on performance than on neuroscientific validation. The claimed interpretability of the learned clusters can be further strengthened.

---

> ### Author Rebuttal · Authors · 2026-03-31
>
> Thank you for the thorough review and constructive comments. We are deeply grateful for acknowledging our method “main strength is its clear and well-motivated idea: learning data-driven functional clusters instead of treating channels independently or relying on fixed anatomical groupings”. Please see below for the point-by-point responses to your comments.
>
> # Reply to Questions
>
> **R to Q1**: Thank you for helpful suggestion. The subscript in Equation (2) should be $\bm{e}^{p}_{i,j}+\bm{e}^{t}_{i}$ . We will correct this in the revised manuscript. Our decision to select sinusoidal positional encoding over learnable embeddings was driven by three primary considerations: (1) Extrapolation Capability -- the ability to handle inputs of arbitrary length, thereby accommodating the varying segment durations found across different datasets during pre-training; (2) Parameter Efficiency -- the elimination of the need for additional parameters, which mitigates the risk of overfitting; (3) Inductive Bias-- the introduction of a smooth positional prior that aligns more closely with the temporal continuity inherent in neural signals.
>
> **R to Q2**: Thank you for helpful suggestion. We acknowledge the importance of providing a neuroscience-based interpretation of clustering results. It is worth noting that while the FCP stage extracts functional connectivity in a data-driven manner, the core objective of NeuroCLUS is to leverage this information to cluster channels into distinct brain region prototypes. Although the model does not explicitly incorporate anatomical priors, our analysis on the Brain Treebank dataset reveals that—when aggregating the attention weights of individual channels across different prototypes—a clear modular distribution emerges, and these modules exhibit anatomical continuity (e.g., frontal electrodes tend to be assigned to one prototype, while temporal electrodes are assigned to another). We further calculated the ratio of intra-cluster distance to inter-cluster distance; the resulting value of 0.1652$\pm$0.0251 quantitatively validates the anatomical continuity of the clustering. Cross-subject analysis indicates that, despite individual variations in the specific assignment of single channels, the spatial topological patterns of the prototypes remain consistent across the vast majority of subjects. This demonstrates that NeuroCLUS can automatically learn functional modules aligned with anatomical structures through a data-driven approach, thereby exhibiting cross-subject generalization. We will incorporate this analysis and the corresponding visualization results into the revised manuscript.
>
> **R to Q3**: Thank you for valuable feedback. Contemporary deep learning models demonstrate significant advantages in handling temporal dependencies. In the context of spontaneous neural activity—such as the Du-IN speech production task—neural signals lack explicit temporal anchors; consequently, traditional machine learning methods struggle to effectively model their dynamic temporal structures, rendering the advantages of deep learning particularly pronounced (SVM: 7.82%, GLM: 5.49%). Conversely, for event-evoked neural activity—such as that found in the Brain Treebank—where characteristic waveforms are already aligned relative to the onset of stimuli, the performance gap between traditional machine learning and deep learning narrows relatively; nevertheless, the original experiments involving BrainBERT still indicate a performance advantage for deep learning. Building upon this foundation, NeuroCLUS further incorporates functional clustering, achieving state-of-the-art results in both categories of tasks. We will supplement this discussion accordingly.
>
> **R to Q4**: Thank you for helpful suggestion. We will include the statistical test results for Figures 2 and 4a in the revised manuscript. Specifically, in Figure 2, with the exception of the Speech task on the Brain Treebank ($p<0.05$), none of the other comparisons reached statistical significance (consistent with the findings of the ablation study); in Figure 4a, all comparisons demonstrated significant differences ($p<0.01$; Section 4.6). We will use paired t-tests (based on 6 random seeds) to annotate the corresponding p-values directly on the figures.
>
> # Reply to Limitations
>
> **R to L1**: Thank you for helpful suggestion. We acknowledge that the SWEC dataset constitutes the dominant portion (approximately 95\%) of our pre-training corpus; however, this fact only serves to render the model's exceptional performance on out-of-distribution datasets—such as MAYO, FNUSA, Brain Treebank, and Du-IN -- even more compelling, thereby conclusively demonstrating NeuroCLUS's cross-dataset generalization capabilities. As more public data becomes available, we look forward to more systematically evaluating the performance of functional clustering-based models across a wider variety of pre-training corpora.

---

> > ### Author Rebuttal · Reviewer_6weS · 2026-04-03
> >
> > Thanks for the clarification in addressing the issue.

---

> > > ### Author Response · Authors · 2026-04-04
> > >
> > > We are grateful to the reviewer for the thorough review and follow-up feedback. We are pleased that our response addressed the issues you raised, and we appreciate you raising the score to 5! We will carefully incorporate all suggested revisions to enhance the clarity and completeness of the paper. Thank you once again for your valuable time and constructive comments.

---

### Official Review · Reviewer_56jX · 2026-03-12

**Soundness:** 2
**Presentation:** 2
**Significance:** 3
**Originality:** 2
**Overall Recommendation:** 3
**Confidence:** 3

**Summary:**

Paper 1345 argues that foundation models for intracranial neural decoding are limited by their inability to learn functionally coherent, region-level representations from channel-level tokenization.
NeuroCLUS is introduced to address this issue.
NeuroCLUS is a foundation model that uses a two-stage pretraining strategy to identify and utilize data-driven functional clusters.
Extensive experiments were conducted on speech perception, speech production, and seizure detection tasks.
NeuroCLUS performs well on these benchmarks, particularly on the Du-IN speech production benchmark.
Thus, functional clustering is presented as a powerful paradigm for building more effective, brain-inspired foundation models for neural decoding.

**Compliance With Llm Reviewing Policy:**

Affirmed.

**Final Justification:**

I find that the authors did a fair job with the rebuttal.
In particular, they addressed the lack of comparison with dictionary learning. However, I think that their approach to dictionary learning is a bit limited, although I am not expert enough to judge they really considered SOTA solutions.

I find the paper still a bit shallow: it basically introduces a clustering step in a complex predictive pipeline. This is fair, but not excited.
My impeession is that the contribution is a bit narrow.

**Key Questions For Authors:**

* Dictionary learning and functional clustering are very similar. However, dictionary learning is more flexible and can fit data better, so it outperforms functional clustering in all cases where I have been able to compare them. This suggests that the authors did not consider strong competitors to their approach.
* It is unclear to me whether "functional localization" is important except in very rare cases, such as presurgical planning.  I wonder if the paper artificially creates constraints that make the proposed NeuroCLUS solution attractive.
* "The data for the five downstream tasks were included in the pre-training corpus." I am curious to see what would happen if this were not done.  There are many situations in which the data for the downstream tasks cannot be part of the pre-training corpus.
* The titles, tick labels, legends, and axis labels in Figures 3 and 4 are illegible.
* It is not mentioned whether the code used by the authors is available.

**Limitations:**

yes

**Strengths And Weaknesses:**

Strengths:

* Experiments are carried out on different tasks and datasets, which is beneficial.
* The proposed methodology seems simple.

Weaknesses:

* The contribution sounds mostly empirical and provides little insight.
* Hyperparameters are not described, e.g., loss coupling.

---

> ### Author Rebuttal · Authors · 2026-03-31
>
> Thank you for the thorough review and constructive comments. We are deeply grateful for acknowledging our method “is simple but beneficial on different tasks and datasets”. Please see below for the point-by-point responses to your comments.
>
> # Reply to Weaknesses
>
> **R to W1**: Thank you for valuable feedback. NeuroCLUS is grounded in the neuroscientific foundation of modular brain function. Since anatomical regions cannot precisely map to function, we explicitly model channel clustering to discover data-driven functional units. This is innovative: previous models stop at channel-level representations, while our FCP stage first discovers functional connections, providing a strong prior for learning brain region-level representations, significantly boosting performance on cognitive tasks (Du-IN, Brain Treebank). This is paradigm innovation with neuroscientific support. We analyzed Brain Treebank: aggregating attention weights across prototypes reveals modular, anatomically continuous patterns (e.g., frontal vs. temporal). The intra-/inter-cluster distance ratio is 0.1852$\pm$0.0251, quantitatively validating anatomical continuity. Cross-subject spatial topologies remain consistent, showing data-driven learning of anatomically aligned functional modules with cross-subject generalization. We will add this analysis and visualizations.
>
> **R to W2**: Thank you for helpful suggestion. We use $L=L_{mse}+L_{clus}$ without extra weighting because both losses have similar numerical scales. Ablation over $\frac{\lambda_{clus}}{\lambda_{mse}}\in\\{0.2, 0.5, 1.0, 2.0, 5.0\\}$ shows stable performance (fluctuation $\leq$2%). We will add this to the appendix.
>
> # Reply to Questions
>
> **R to Q1**: Thank you for helpful suggestion. Dictionary learning methods (e.g., LaBraM, Du-IN) learn channel-level discrete codebooks; each channel independently retrieves a codeword, without compressing multiple channels into a region-level representation. NeuroCLUS first identifies each channel’s functional group (prototype) then aggregates channels within the same group into a region-level prototype - explicit “channel$\rightarrow$region” compression. We compared against LaBraM (channel-level dictionary) and H2DiLR (region-level dictionary; Du-IN also belongs this category); NeuroCLUS outperforms all.
>
> | Methods | SWEC (Kappa) | SWEC (F1) | MAYO (F1) | FNUSA (F1) | Pitch | Volumn | Onset | Speech | Du-IN |
> | --- | --- | --- | --- | --- | --- | --- | --- | --- | --- |
> | LaBraM | 0.51$\pm$0.02 | 0.47$\pm$0.02 | 0.33$\pm$0.02 | 0.44$\pm$0.03 | 0.72$\pm$0.02 | 0.86$\pm$0.02 | 0.89$\pm$0.02 | 0.94$\pm$0.01 | 12.82$\pm$2.51 |
> | H2DiLR | 0.52$\pm$0.02 | 0.47$\pm$0.02 | 0.30$\pm$0.02 | 0.43$\pm$0.03 | 0.74$\pm$0.02 | 0.87$\pm$0.02 | 0.88$\pm$0.02 | 0.90$\pm$0.02 | 23.92$\pm$3.79 |
>
> **R to Q2**: Thank you for valuable feedback. In this paper, "functional localization" does not refer to clinical surgical localization, but rather to the model's automatic identification of which channels belong to the same functional group and aggregating them into region-level representations based on pre-defined prototypes (Figure 1b & Section 2.2). This is the key difference between this paper and two existing models: (1) Channel-level models (such as Brant) stop at channel-level representations (2) Brain-level models (such as PopT) only support global aggregation (Section 2.2). We achieved SOTA results on three completely different tasks, strictly adhering to existing evaluation criteria -- particularly on the cognitively complex Du-IN dataset (65.92\%). This demonstrates that functional clustering is broadly important in general neural decoding, rather than an artificial constraint. We did not alter any benchmark evaluation protocols, and all comparisons are fair and transparent.
>
> **R to Q3**: Thank you for valuable feedback. We would like to subtly point out that this issue has been specifically discussed and verified in Section 4.4 and Figure 2 of the paper. Specifically, we conducted ablation experiments by removing the MAYO & FNUSA, Brain Treebank, and Du-IN datasets from the pre-training corpus, and the results showed that the model's performance on downstream tasks was not significantly affected. This fully demonstrates that NeuroCLUS learns general neural representations, rather than relying on specific downstream data appearing in the pre-training corpus. Therefore, even when the target task data cannot be included in the pre-training corpus in practical applications, the model can still maintain strong generalization ability. We will further emphasize this conclusion in the revised version.
>
> **R to Q4**: Thank you for helpful suggestion. We will regenerate Figs.3&4 with clear vector fonts.
>
> **R to Q5**: Thank you for valuable feedback. Definitely, we will make our codes and model weights publicly available based on publication.

---

> > ### Author Rebuttal · Reviewer_56jX · 2026-04-01
> >
> > I appreciate the authors' efforts to address my comments.
> > I think that the comparison with Dictionary learning s a bit shallow, and I would like more clues to understand the results.
> > Regarding Q3 I take the authors' reponse, but I find the paper presentation quite awkward regarding this point.

---

> > > ### Author Response · Authors · 2026-04-02
> > >
> > > Thank you for the thorough review and follow-up comments. Please see below for the point-by-point responses to your comments.
> > >
> > > # Reply to Supp-Questions
> > >
> > > **R to SQ1**: Thank you for valuable feedback. We understand that the reviewers are seeking further insights to explain the discrepancies in the results. First, let us clarify the core objective of these dictionary learning methods: whether it be LaBraM [1] (channel-level), Du-IN [2] / H2DiLR [3] (region-level), they all construct discrete codebooks via self-supervision to provide more abstract self-supervisory signals for the Masked Autoencoding (MAE) stage. However, they suffer from the following key limitations:
> > >  - LaBraM (Channel-level Dictionary): It treats all channels equally and is unable to form region-level representations; consequently, it performs extremely poorly (12.82\%) on the Du-IN speech decoding task, which requires high-level cognition, although the performance gap narrows on the Brain Treebank due to the relative simplicity of the tasks involved.
> > >  - Du-IN / H2DiLR (Region-level Dictionary): Although these methods encourage region-level representations, they do not constitute foundational models—they are unable to effectively leverage cross-subject data to construct a unified representation space, and their performance remains constrained by within-subject training. In the context of seizure detection tasks—where the specific brain regions exhibiting abnormal discharges cannot be determined *a priori*—their intended region-level representations are effectively forced to degrade into brain-level representations. Furthermore, due to inconsistencies in the number of recording channels across subjects, actual evaluation still relies on within-subject data, meaning the approach does not truly achieve zero-shot generalization. We will elaborate on this point in the experimental section. This constitutes the most significant drawback of region-level dictionary learning methods regarding their clinical generalizability.
> > >
> > > In comparison, the advantages of NeuroCLUS lie in the following:
> > > - **Compared to LaBraM & Du-IN**, it constructs a unified functional clustering space through cross-subject pre-training, achiving strong performance on cognitive decoding tasks (e.g., Du-IN and Brain Treebank);
> > > - **Compared to LaBraM**, its clustering loss simultaneously maximizes intra-cluster compactness and inter-cluster separability—a design that aligns closely with the pathological characteristics of epilepsy, specifically the "abnormal functional discharge of brain regions";
> > > - **Compared to Du-IN**, it requires no predefined brain regions, instead employing a data-driven approach to automatically discover functional groups, thereby achieving significantly superior performance in seizure detection.
> > >
> > > Furthermore, reviewer might perceive PopT as a method akin to functional clustering. We wish to clarify that PopT decouples spatiotemporal modeling; consequently, its performance falls short of that achieved by LaBraM and CBraMod [4] on TUAB EEG dataset [1], which simultaneously optimize spatiotemporal aspects. Notably, CBraMod does not employ dictionary learning; instead, it surpasses LaBraM through architectural innovation. A recent systematic study [5] published at ICLR also revealed that, in some instances, performance gains derived from architectural optimization can exceed those yielded by dictionary learning objectives. NeuroCLUS jointly optimizes spatiotemporal dependencies and functional clustering within a unified architecture—specifically implemented in a manner similar to LaBraM and CBraMod—thereby **achieving a superior balance between cognitive and clinical decoding performance**.
> > >
> > > **R to SQ2**: Thank you for helpful suggestion. We fully agree with the reviewer's assessment that the phrasing in Section 4.4 is "awkward." In the revised manuscript, we will completely rewrite this section: we will begin by directly stating that "the inclusion of downstream task data during pre-training does not impact performance," and will supplement this with clearer figure captions and textual guidance to prevent any potential misunderstandings. We will incorporate these commitments into the final revised manuscript.
> > >
> > > # References
> > >
> > > [1] Jiang W B, Zhao L M, Lu B L. Large brain model for learning generic representations with tremendous EEG data in BCI, 2024.
> > >
> > > [2] Zheng H, Wang H T, Jiang W B, et al. Du-IN: Discrete units-guided mask modeling for decoding speech from intracranial neural signals, 2024.
> > >
> > > [3] Wu D, Li S, Feng C, et al. Towards homogeneous lexical tone decoding from heterogeneous intracranial recordings, 2024.
> > >
> > > [4] Wang J, Zhao S, Luo Z, et al. Cbramod: A criss-cross brain foundation model for eeg decoding, 2024.
> > >
> > > [5] Yang L, Sun Q, Li A, et al. Are EEG foundation models worth it? comparative evaluation with traditional decoders in diverse BCI tasks, 2026.

---

### Decision · Program_Chairs · 2026-04-30

**Decision:**

Accept (regular)

**Comment:**

NeuroCLUS is a foundation-style model for intracranial EEG recordings that rejects two common tokenization extremes: treating every electrode as an independent token, or collapsing the whole brain into one global token. Instead, it learns data-driven functional clusters of channels. The training recipe is two-stage: first estimate functional connectivity between channels to build a graph prior, then use that prior to soft-assign channels to learnable prototype tokens that a transformer backbone consumes. The paper evaluates on speech-related tasks, seizure detection, and other intracranial benchmarks, reporting competitive or state-of-the-art numbers against recent baselines.

The functional-clustering idea is easy to motivate for iEEG where spatial sampling is irregular and anatomy alone is a poor proxy for function. Reviewers highlighted the empirical breadth (multiple datasets and task families) and the fact that comparisons included modern baselines (e.g. MVPFormer, BaRISTA) rather than only classical pipelines. The paper also tries to connect learned clusters to interpretability, which matters for clinical-adjacent decoding. The ablation section in the original submission was already considered useful.

The first round raised many practical questions: which public datasets constitute pretraining, how baselines were trained or checkpointed, how BaRISTA is used when anatomical coordinates are missing, and how connectivity is extracted from attention for clustering. Reviewers also asked for clearer differentiation from related work (PopT, GAFormer), more ablations, and honest limits of coarse groupings. The rebuttal was long and detailed: same-pretraining-data retraining for baselines, parameter counts, EMA details, ablations on Du-IN and related tasks, and extra analyses on Brain Treebank prototypes (distance ratios, cross-subject consistency). Several reviewers marked issues fully resolved and increased scores.

One reviewer stayed at a borderline positive score: they credited dictionary-learning comparisons but still felt the contribution reads somewhat like a strong engineering pipeline, and wanted sharper narrative on pretraining overlap with downstream data. Authors replied again with clarifications and commitments to rewrite awkward sections. Three reviewers ended clearly positive after rebuttal (including two who moved to strong accept or accept). One reviewer remained partially satisfied but did not overturn the overall direction.

Fully or largely closed for most reviewers: pretraining data transparency, baseline training parity, connectivity extraction details, ablations on key tasks, and citations to related work that were missing in round one. Still partially open for at least one reviewer: how much the paper is “conceptual novelty” versus a well-engineered combination, and how cleanly the presentation separates pretraining overlap from downstream evaluation. The authors committed to rewrites rather than claiming new theorems.

Some baseline caveats (e.g. BaRISTA without full anatomical embeddings on some cohorts) need careful wording in the camera-ready, and the “one skeptical reviewer” concerns about presentation should be honored in revision.

Despite these remaining issues, the panel consensus is that the idea is well motivated and the empirical package is strong enough for the conference. Given these strengths, we accept.